

# Greenhouse gas emissions from fen soils used for forage
# production in northern Germany
**Arne Poyda[1,*], Thorsten Reinsch[1], Christof Kluß[1], Ralf Loges[1,] Friedhelm Taube[1]**
[1]Institute for Crop Science and Plant Breeding, Grass and Forage Science/Organic Agriculture, Kiel
University, Hermann-Rodewald-Str. 9, 24118 Kiel, Germany
*now at: Institute of Soil Science and Land Evaluation, Biogeophysics, Hohenheim University, Emil-
Wolff-Str. 27, 70593 Stuttgart, Germany
*Correspondence to*: Arne Poyda (a.poyda@uni-hohenheim.de)
## Abstract
A large share of peatlands in northwest Germany is drained for agricultural purposes, thereby emitting
high amounts of greenhouse gases (GHG). In order to quantify the climatic impact of fen soils in dairy
farming systems of northern Germany, GHG exchange and forage yield were determined on four
experimental sites which differed in terms of management and drainage intensity: a) rewetted and
unutilized grassland (UG), b) intensive and 'wet' grassland (GW), c) intensive and 'moist' grassland
(GM) and d) arable forage cropping (AR). Net ecosystem exchange (NEE) of $CO_2$ and fluxes of $CH_4$
and $N_2O$ were measured using closed manual chambers. $CH_4$ fluxes were significantly affected by
groundwater level (GWL) and soil temperature, whereas $N_2O$ fluxes showed a significant relation to the
amount of nitrate in top soil. Annual balances of all three gases, as well as the global warming potential
(GWP), were significantly correlated to mean annual GWL. Two-year mean GWP, combined from
$CO_2$-C-equivalents of NEE, $CH_4$ and $N_2O$ emissions, as well as C input (slurry) and C output (harvest),
was 3.8, 11.7, 17.7 and 17.3 Mg $CO_2$-C-eq ha$^{-1}$ a$^{-1}$ for sites UG, GW, GM and AR, respectively
(standard error (SE) 2.8, 1.2, 1.8, 2.6). Yield related emissions for the three agricultural sites were 201,
248 and 269 kg $CO_2$-C-eq (GJ net energy lactation (NEL))$^{-1}$ for sites GW, GM and AR, respectively
(SE 17, 9, 19). The carbon footprint of agricultural commodities grown on fen soils depended on long-
term drainage intensity rather than type of management, but management and climate strongly
influenced interannual on-site variability. However, arable forage production revealed a high
uncertainty of yield and therefore was an unsuitable land use option. Lowest yield related GHG
emissions were achieved by a three-cut system of productive grassland swards in combination with a
high GWL (long-term mean $\leq$ 20 cm below the surface).

## 1   Introduction

Natural peatland ecosystems act as long-term carbon (C) sinks as C in plant residues accumulates due to anoxic conditions and thus incomplete decomposition (Joosten & Clarke, 2002). Globally, the amount of C stored in peatlands is about 446 Pg (2 Pg in German peatlands) (Joosten, 2009), which is 24 % higher compared to the number of 359 Pg C stored in global forest vegetation, given by Dixon et al. (1994). The drainage of peatlands causes aerobic soil conditions, leading to accelerated mineralization of the soil organic matter (SOM) and an increased release of C and nitrogen (N) (Höper, 2002). Therefore, the natural sink for C and N is turned into a net source, converting drained peatlands to significant emitters of the greenhouse gases carbon dioxide ($CO_2$) and nitrous oxide ($N_2O$) (Kasimir-Klemedtsson et al., 1997; Maljanen et al., 2003b, 2010). Simultaneously, the methane ($CH_4$) emissions occurring under natural conditions are reduced to negligible levels (Roulet et al., 1993; van den Pol-van Dasselaar et al., 1997; Maljanen et al., 2003a).

In Germany, peatlands cover around 1.67 million ha (Joosten, 2009), which corresponds to 4.7 % of the land area. Roughly 65 % of these peatlands are minerotrophic fens (Grosse-Brauckmann, 1997) and around 70 % is utilized for agricultural purposes (Röder & Osterburg, 2012). Peatland rich regions, as particularly northwest (NW) Germany (Lower Saxony, Schleswig-Holstein), show high shares of forage production and livestock units per ha of utilized agricultural area, which is attributed to a concentration of dairy farming (Röder & Osterburg, 2012). Consequently, there is a high demand for intensive forage production to ensure the supply of a high quality fodder. These management and cultivation practices require an intensive drainage and fertilization, leading to a continually increasing pressure on the utilization of German peatlands. The relevance of agricultural utilized peatlands for the national GHG budget is highlighted as only 5 % of the utilized agricultural area (Röder et al., 2011) but 50 % of the GHG emissions from agricultural soils (41.3 of 82.7 Tg $CO_2$-equivalents ($CO_2$-eq)) are attributed to peatlands drained for agriculture (UBA, 2014).

Restoration of cultivated organic soils has one of the greatest GHG mitigation potentials in agriculture (Smith et al., 2008). The reestablishment of the natural peatland functioning can only be achieved by abandoning the drainage based utilization, accompanied with a rewetting to natural hydrological conditions (Gorham & Rochefort, 2003; Höper et al., 2008; Zak et al., 2011). However, removing land from production provides maximum GHG mitigation, but might be rather an option for marginal lands than for regions with a high agricultural production value (Robertson et al., 2000). In those regions, it becomes fundamental to identify mitigation options that reduce GHG emissions without a distinct



1. reduction of the agricultural productivity (Smith et al., 2008). Furthermore, the objective of climate protection measures for these areas should focus on resource use efficiency, i.e. minimizing GHG emissions per unit of product instead of unit area (Oenema et al., 2014). Here, we will focus on the net exchange of the three biogenic trace gases $CO_2$, $CH_4$ and $N_2O$ from fen soils in an intensive dairy farming region of northern Germany (Schleswig-Holstein) and relate their annual budgets to forage energy yield (net energy lactation, NEL) of the specific sites.

7. There are several publications about the climatic relevance of peatlands and their corresponding emission factors (Byrne et al., 2004; Alm et al., 2007a; Drösler et al., 2008; Oleszczuk et al., 2008; Couwenberg, 2009b; Maljanen et al., 2010). In recent years, advanced information about the GHG fluxes from German peatlands is emerging (Drösler, 2005; Couwenberg, 2011; Beetz et al., 2013; Beyer & Höper, 2014; Leiber-Sauheitl et al., 2014). Nevertheless, GHG data for agricultural managed fen soils in northern Germany is lacking and their function for forage production has not been considered in calculations about GHG mitigation. Therefore, the recommended strategy for GHG reductions from drained peatlands is the rewetting to natural conditions or extensification (Couwenberg et al., 2011; Beetz et al., 2013). However, in terms of reducing GHG emissions per unit forage produced, Renger et al. (2002) and Regina et al. (2014) report consistently that an average groundwater table of 30 cm below the soil surface enables high yielding grass cultivation and reduces the GHG emissions for a minimum of 40 %.

19. This study provides a full GHG balance as well as forage yields of fen soils in northern Germany in an intensive dairy farming region with different management strategies: a) rewetted and unutilized grassland (UG), b) intensive grassland 'wet' (GW), c) intensive grassland 'moist', (GM) and d) arable forage production (AR) and the assumptions that:

23. (i) rewetting leads to a decrease in $CO_2$ and $N_2O$ emissions but an increase in $CH_4$ emissions,

24. (ii) the GHG balances and C losses increase with land use intensity in the order UG > GW > GM > AR,

26. (iii) product related GHG emissions are higher for arable forage cropping on organic soils compared to grassland utilization,

28. (iv) wet but intensive grassland utilization (site GW) realizes lowest product related GHG emissions.



## 2 Material and Methods

### 2.1 Study area

The study was conducted in a huge lowland area of Schleswig-Holstein, the most northern state of Germany, at 54°21' N and 9°24' E. The long-term (1981 – 2010) mean annual temperature in this region is 8.7 °C and mean annual precipitation is 861 mm (Deutscher Wetterdienst (DWD), 2011). The region was shaped by meltwater at the end of the last ice age (Weichsel glacial stage) that flowed through the valleys originated by the previous ice age (Saale glacial stage). Thereby, river systems were formed and as a result of sea level and groundwater rise, deep fen soils developed that grew up to peat bogs at some locations (Blume & Brümmer, 1986). Since several centuries the area has been drained for agricultural utilization. Traditionally, the fen soils of the study area have been used as grasslands for forage production in dairy farms. In the past two decades about 15,000 ha of the region have been allocated for nature conservation purposes. In these areas the water levels were permanently raised and the agricultural utilization was extensified or abandoned (Rohman et al., 2008).

As a result of the ground level elevation as well as the status of the drainage system, the study area is irregularly drained, resulting in highly variable groundwater levels and thus intensity of peat degradation. According to these conditions, four sites were selected representing typical land use and drainage scenarios in this region. A rewetted and unutilized grassland site (UG) was chosen to evaluate the situation without agricultural activities. This site is located in a nature reserve area and was rewetted in 1991. There has been no utilization since 1998 and no fertilization since the rewetting. The vegetation of site UG is typical for wet and nutrient rich fallows, with a few dominant and productive species (Timmermann et al., 2006; Schrautzer et al., 2013). In contrast, the vegetation composition of the utilized grasslands (grassland 'wet', GW and grassland 'moist', GM) is dominated by species typical for intensively managed temperate grasslands (Table 1). The arable site (AR) was used as permanent grassland until conversion to silage maize production in 2007. In 2012, the cultivation changed to production of whole crop silage from spring barley and from spring wheat with undersown grass in 2013. The soil types of all sites are classified as *Histosols* according to FAO (2006).

The utilized grassland sites are fertilized with slurry from dairy cattle. Typically, this is conducted shortly before the beginning of the growing season in a range of 20 – 30 m³ ha$^{-1}$ and subsequently after cutting events in a smaller range of 10 – 15 m³ ha$^{-1}$ if another cutting is designated. The arable site received 35 and 18 m³ ha$^{-1}$ of cattle slurry in 2011 and 2012, respectively. The slurry was deployed and incorporated into the top soil immediately before the sowing of the crops. In 2013, no slurry was applied. Additionally, the agricultural sites received mineral N fertilizers around the same dates as the





slurry application, which occurs mostly in the form of calcium ammonium nitrate (CAN), containing 27
% of N. The total amounts of applied fertilizer N are displayed in Table 1.

## 2.2   Site characteristics

Air temperature, precipitation and photosynthetically active radiation (PAR) were measured at a climate
station on site GW. When missing data occurred due to technical problems, data from a meteorological
station of the DWD, located about 5 km from the sites, was used for gap filling. Soil temperatures in 5,
10 and 15 cm depth of each site were continuously recorded every hour by soil temperature loggers
(SL52T, IMEC, Heilbronn, Germany).

### 2.2.1  Groundwater levels

For continuous monitoring of groundwater levels (GWLs), four perforated PVC tubes (d = 3 cm, l =
120 cm) were installed on each site in pairs at 5 and 15 m from the next drainage ditch. GWLs were
recorded manually during every gas flux measurement campaign, leading to a minimum of one GWL
record per week. For the calculation of mean annual GWLs, the recorded GWLs were linear
interpolated to obtain daily values and to avoid overestimation of periods with more frequent
measurements.

### 2.2.2  Soil properties

For monitoring of soil mineral N status, soil samples were taken fortnightly with a soil auger at a depth
of 0 – 20 cm on each site. Nitrogen was extracted with 0.01 M $CaCl_2$ (VDLUFA, 1997) and the
concentrations of nitrate ($NO_3^-$) and ammonium ($NH_4^+$) of the extractions were analyzed
photometrically with a dual channel continuous flow analyzer (San$^{++}$, Skalar Analytical B.V., Breda,
The Netherlands). Mineral N stocks per ha were calculated using the bulk density of the relevant sites.
Bulk density was determined for the depths 5, 15, 25 and 45 cm according to DIN ISO 11272 (HBU,
1998). The gravimetric water content of soil samples was estimated by oven drying at 105 °C. To
calculate the contents and amounts of $C_{org}$ and $N_{tot}$ of each site, soil sampling was conducted twice a
year at soil depths of 0 – 30, 30 – 60 and 60 – 90 cm. After oven drying (40 °C), samples were analyzed
with an elemental analyzer (Vario Max CN, Elementar, Hanau, Germany). The soil pH was determined
before and after the study period in 2011 and 2014 according to VDLUFA (1991).



### 2.2.3 Herbage yield and forage quality

To quantify the herbage yields, the above ground biomass (AGB) was cut shortly before harvest on three randomly selected spots with 0.25 m² at a height of 5 cm. The dry matter content of plants was determined after oven drying at 60 °C for 48 h. Subsequently, the material was grinded using a centrifugal mill equipped with a 1 mm sieve (Cyclotech mill, Tecator, Foss, Hillerød, Denmark). Forage quality parameters were estimated by near infrared reflectance spectroscopy (NIRS) (Baker & Barnes, 1990). Therefore, each sample was scanned with a NIR-System 5000 monochromator (FOSS, Silver Spring, USA). The NIRS calibrations were based on a sample pool selected to represent the entire spectral and chemical variability for which N concentrations were directly measured with an elemental analyzer (Vario Max CN, Elementar, Hanau, Germany). Net energy lactation (NEL) as the feed energy content available for maintenance and milk production was estimated as a function of metabolizable energy (ME) and crude ash content (Weißbach et al., 1996), whereas ME was calculated from the contents of enzyme soluble organic matter, crude ash, crude fat and acid detergent fiber according to GfE (2008).

### 2.3 Determination of GHG fluxes and balances

### 2.3.1 Flux measurements

$CH_4$ and $N_2O$ fluxes were measured from April 2011 to March 2014 using closed manual chambers (Hutchinson & Mosier, 1981). Measurements were conducted weekly and in addition shortly after management practices like fertilization or tillage. At each site, eight PVC collars (d = 60 cm, h = 15 cm) were inserted 10 cm into the soil one week before the measurements started. To display gas fluxes for different GWLs at the same time, four collars were placed at 5 and 15 m from the next drainage ditch, respectively. When sites were harvested, the vegetation was removed from the collars. Site preparation measures were conducted in spring and the collars were shifted afterwards to obtain representative conditions. On site UG, a boardwalk was installed due to wet soil conditions and to avoid disturbances around the collars. For gas flux measurements, opaque PVC chambers (h = 35 cm, V = 0.1 m³) were used and chamber air samples were collected with a 30 ml syringe and stored in 12 ml pre-evacuated septum capped vials (Labco, High Wycombe, UK) (Glatzel & Well, 2008) 0, 15 and 30 min after chamber closure. Sampling was conducted between 09:00 and 12:00 to capture mean daily fluxes (Velthof & Oenema, 1995a; Petersen et al., 2012; van der Weerden et al., 2013). The samples were analyzed for concentrations of $CH_4$, $N_2O$ and $CO_2$ with a gas chromatograph (7890a, Agilent Technology Inc., Santa Clara, CA, USA) equipped with a flame ionization detector (FID), electron capture detector (ECD) and thermal conductivity detector (TCD). Calibration of the gas chromatograph





was performed with a minimum of three certified gas standards. Samples were injected using an
autosampler (222 XL, Gilson Inc., Middleton, WI, USA). Data processing was conducted with the
software *Chem Station* (Version B.01.04, Agilent Technology Inc., Santa Clara, CA, USA).
The $CO_2$ exchange was determined according to the method of Drösler (2005). Elsgaard et al. (2012),
Beetz et al. (2013) and Leiber-Sauheitl et al. (2014) present similar approaches. Here, static chambers
with a diameter of 61 cm and a height of 35 were used. On each site three PVC collars were installed.
Measurement campaigns were conducted during the period March 2012 until April 2014 in intervals of
3 to 5 weeks. When harvest of the agricultural sites took place, the vegetation was removed from the
collars and additional $CO_2$ measurements were carried out few days after harvest. In total, the $CO_2$
exchange was measured on 21, 28, 30 and 32 days at site UG, GW, GM and AR, respectively.
Transparent and opaque chambers were used to measure the net ecosystem exchange (NEE) and the
ecosystem respiration ($R_{ECO}$), respectively. The chambers were connected to an infrared gas analyzer
(LI-820, LI-COR Biosciences, Lincoln, NE, USA) and a data logger (CR 1000, Campbell Scientific,
Logan, UT, USA). $CO_2$ concentration inside the chamber, temperature inside and outside the chamber
and PAR outside the chamber were recorded every 5 s. Chambers were equipped with a fan to ensure
homogenization of the atmosphere inside the chamber headspace. When the vegetation was higher than
the chambers, extensions (h = 35 cm) were used. Measurement campaigns were conducted from sunrise
until afternoon to comprise the whole daily range of PAR and soil temperature. Maximum enclosure
times were 120 s for NEE and 300 s for $R_{ECO}$ measurements. Quality criteria for $CO_2$ measurements
were changes of chamber temperature of more than 1.5 °C and a standard deviation of PAR more than
10 % of average PAR. Measurements that exceeded these threshold values were discarded.

## 2.3.2 Flux calculations

Trace gas fluxes were calculated using linear regression for the change of gas concentration over time
as it has been described in several other studies (e.g. Flessa et al., 1998; Chatskikh et al., 2008; Beetz et
al., 2013). Since effects of temperature and pressure inside the chamber induce only minor uncertainties
to the measured fluxes (Levy et al., 2011), these variables are often neglected in flux calculations
(Chatskikh et al., 2008). However, to quantify the uncertainty in calculated $CO_2$ fluxes caused by a
varying density of air as a function of temperature, $CO_2$ fluxes (n = 5546) were corrected for the mean
temperature inside the chamber and compared to the uncorrected fluxes. On average, temperature
correction reduced calculated fluxes by 6 % with a maximum reduction of 12 % at a very high
temperature of 38 °C. As temperature was not measured inside the chambers for $CH_4$ and $N_2O$ flux
measurements, the uncorrected $CO_2$ fluxes were used for further analyses to ensure methodological
consistency.



For $CH_4$ and $N_2O$, fluxes were accepted when the coefficient of determination ($R^2$) of the linear
regression was $\geq 0.9$ to ensure a high accuracy of measured fluxes. Measurements with $R^2 < 0.9$
occurred mainly when chamber concentrations were near ambient and the corresponding fluxes were
assumed to be 0. $CO_2$ concentrations of the gas samples were used as control to identify erroneous $CH_4$
and $N_2O$ values. If the $CO_2$ concentration of a sample was not plausible (i.e. smaller than previous), the
fluxes of $CH_4$ and $N_2O$ were discarded from the dataset (Leiber-Sauheitl et al. 2014). For NEE and $R_{ECO}$
measurements, all fluxes with plausible concentration changes over time were accepted, irrespective of
flux magnitude and the $R^2$ of linear regression (Alm et al., 2007b; Leiber-Sauheitl et al., 2014). To
avoid underestimation of $CO_2$ exchange by a diminishing concentration gradient between chamber
headspace and soil or plant, and thus decreasing fluxes (Davidson et al., 2002), only the part of linear
concentration change was used for flux calculation, which could be only 30 s for NEE measurements
with highly productive vegetation and high PAR.

### 2.3.3 $CO_2$ modelling

$R_{ECO}$ was estimated using a temperature-dependent flux model according to Lloyd & Taylor (1994):

$$R_{ECO} = R_{ref} * \exp\left[ E_0 * \left( \frac{1}{T_{ref} - T_0} - \frac{1}{T - T_0} \right) \right] \qquad (1)$$

where $R_{ECO}$ is the measured ecosystem respiration (g $CO_2$-C m$^{-2}$ h$^{-1}$), $R_{ref}$ is the respiration at reference
temperature (g $CO_2$-C m$^{-2}$ h$^{-1}$), $E_0$ is an activation-like parameter (K), $T_{ref}$ is the reference temperature
(283.15 K), $T_0$ is the temperature constant for the start of biological processes (227.13 K), and T is the
temperature with the best fit to the data of one measurement campaign. This could be either soil
temperature in 5 cm depth at the corresponding site or the air temperature from the weather station at
site GW. For modelling $R_{ECO}$, $R_{ref}$ and $E_0$ were fitted plot based for each measurement campaign with
soil or air temperature, depending on the level of significance. If neither soil temperature nor air
temperature gave a significant relation to $R_{ECO}$ of a measurement campaign, the data was pooled with
that of one or two adjacent campaigns to obtain significant parameters for the $R_{ECO}$ model (Beetz et al.,
2013). However, for site UG it was in some cases not possible to calculate significant parameters.
Therefore, the dataset was separated into growing season and non-growing season according to Janssens
(2010) and all measurement campaigns of a season were pooled. By this approach, the temporal
resolution of the model was decreased, but the range of temperatures for which the model is valid, was
greatly increased. Nevertheless, for the agricultural sites it was necessary to consider the phenological
development of the plants and especially the effect of harvest in the model. When fitting the model per
campaign, the temperature range can be very narrow, which may lead to severe overestimations by the
$R_{ECO}$ model if the slope of regression is high and the temperature is above of the observed range.



Therefore, the highest measured value of the corresponding campaign was set as a threshold for
maximum $R_{ECO}$. Every modelled value exceeding that threshold was recessed. The fitted parameters $R_{ref}$
and $E_0$ were linear interpolated between the campaigns and $R_{ECO}$ was modelled on an hourly basis using
the corresponding temperature. To calculate GPP, the modelled $R_{ECO}$ at the time of NEE measurements
was subtracted from the measured NEE value.
GPP was modelled with PAR as input variable using the rectangular hyperbola of Michaelis & Menten

7 (1913):

$$GPP = \frac{(GP_{max} * \alpha * PAR)}{(GP_{max} + \alpha * PAR)} \qquad (2)$$

where GPP is the calculated gross primary production (g $CO_2$-C m$^{-2}$ h$^{-1}$), $GP_{max}$ is the limit of carbon
fixation for infinite PAR (g $CO_2$-C m$^{-2}$ h$^{-1}$), $\alpha$ is the initial slope of the regression curve or light use
efficiency ((g $CO_2$-C m$^{-2}$ h$^{-1}$) (µmol m$^{-2}$ s$^{-1}$)$^{-1}$) and PAR is the average photon flux density of
photosynthetically active radiation (µmol m$^{-2}$ s$^{-1}$) that was determined during the NEE measurement by
a quantum sensor (SKP 215, Skye Instruments, Llandrindod Wells, UK). PAR was corrected by a factor
of 0.92 as an absorption of 8 % by the transparent chambers was identified by own examinations. $GP_{max}$
and $\alpha$ were fitted plot based for each measurement campaign and linear interpolated between the
campaigns, assuming a consistent development of vegetation. However, as the plant biomass is
harvested, $CO_2$ uptake is interrupted immediately. Therefore, the parameters of the preceding
measurement campaign, which was conducted only few days before harvest, were used until the cutting
and then set back to 0. The subsequent campaign was conducted within one week after the cutting to
capture the $CO_2$ exchange of the recently harvested plants. GPP was modelled on an hourly basis using
measured PAR from the weather station at site GW.
**2.3.4 GHG and C balances**
As the net ecosystem exchange (NEE) of $CO_2$ is the balance of $CO_2$ uptake by plants (GPP) and the
autotrophic and heterotrophic respiration of plants and soil ($R_{ECO}$) (Chapin et al., 2006), NEE was
calculated on an hourly basis as the sum of Eqs. (1) and (2):
$NEE = GPP + R_{ECO}$ \qquad\qquad (3)
For further processing, GPP, $R_{ECO}$ and NEE were calculated per hectare and summed up to daily values
(kg $CO_2$-C ha$^{-1}$ d$^{-1}$). The site specific annual balances of the three components were calculated as the
average of the 365-days sums of the three replicates. Annual $CH_4$ and $N_2O$ balances were determined by
plot based linear interpolation between the measurement days and summation of daily values. Site
specific balances were calculated as average of the eight replicates. The global warming potential



(GWP) of a specific site indicates to which magnitude it contributes to global warming, based on the
GHG balance for a certain period. GWP was calculated using the IPCC (2007) radiative forcing factors
of the individual gases for a time horizon of 100 years. These are 25 for $CH_4$ and 298 for $N_2O$ related to
$CO_2$ ($CO_2$-equivalents ($CO_2$-eq)). Additionally, anthropogenic C inputs and losses via slurry application
and harvest were calculated as $CO_2$-eq and included in the GWP (Beetz et al., 2013). Using the balances
of $CO_2$-C and $CH_4$-C as well as the C import via slurry and C export via biomass harvest, the net
ecosystem carbon balance (NECB) was calculated per site and year. For all C and GHG fluxes and
balances, the atmospheric sign convention was applied, where all losses from the atmosphere into the
ecosystem (site) are displayed as negative (the ecosystem acts as a sink) and all enrichments in the
atmosphere are displayed as positive (the ecosystem acts as a source). This convention is transferred to
the non-atmospheric fluxes like slurry application (negative) and biomass harvest (positive). GHG and
carbon balances were calculated for the periods April 2012 – March 2013 and April 2013 – March

13  2014.

**2.4  Statistical analyses**
The statistical software R (2014) was used to evaluate the data. Evaluation started with the definition of
an appropriate statistical mixed model (Laird & Ware, 1982; Verbeke & Molenberghs, 2000). The data
were assumed to be normally distributed and heteroscedastic due to the different sites and measurement
periods. These assumptions are based on a graphical residual analysis. The statistical model included
the site as a fixed factor. For daily $CH_4$ fluxes, GWL and soil temperature in 5 cm were modelled as
covariates, whereas for $N_2O$ fluxes, the amount of nitrate in 0 – 20 cm soil depth was used. The year
was regarded as a random factor. Also, the correlations of the measurement values due to the day of
sampling were taken into account. Based on this model, an analysis of covariance (ANCOVA) was
conducted to test for significant influences of the covariates.
For balances of $CH_4$, $N_2O$ and $CO_2$, as well as for the GWP, NECB and product related GHG
emissions, a mixed model with the site as fixed factor and the year as random factor was defined in each
case. Heteroscedasticity was modelled due to the different sites and measurement periods. An analysis
of variance (ANOVA) was conducted to identify significant differences between the sites. For the yields
of DM, C, N and NEL, the model was amplified by the year as a fixed factor instead of random factor.
Furthermore, multiple contrast tests (Bretz et al., 2011) were conducted in order to identify significant
differences between sites and years, respectively.
To evaluate the influence of GWL on the different trace gas balances and the total GWP, NECB and
product related GHG emissions, mean annual GWL was added as a fixed factor to the model used for
the *t*-test. This model was calculated with and without the interaction term of site and GWL, as well as




irrespective of the different sites. These three model types were compared referring to their Akaike
information criterion (AIC) (Akaike, 1974) to assess which model gives the best estimate for the
relation between GWL and the corresponding variable. For $CH_4$ and $N_2O$ balances, this procedure was
conducted for mean annual GWL and, in terms of $N_2O$, for mean annual soil nitrate.
For uncertainty analysis of the $CO_2$ model, a Monte Carlo simulation was conducted for each
measurement plot and site. Therefore, model parameters with the same variation as the original values
were randomly calculated for every measurement campaign or pooled dataset and new regressions with
temperature ($R_{ECO}$) and PAR (GPP) were fitted. Only regressions with realistic parameters were
accepted ($E_0$ and $R_{ref} \geq 0$, $\alpha$ and $GP_{max} \leq 0$). This procedure was conducted 10,000 times, thus, 10,000
different model outputs for $R_{ECO}$, GPP and NEE were obtained. The variation of these randomly
calculated model outputs represents the uncertainty that is caused by the chamber measurements and by
the fitting and linear interpolation of different numbers of measurement campaigns per plot and year.
Since this procedure is conducted for each plot, the uncertainty can be calculated as the sum of mean
variance of the three plots per site and the variance resulting from averaging the three replicates. This
uncertainty was used for comparison of means obtained by the original simulation. Leiber-Sauheitl et al.
(2014) present a similar approach.
**3  Results**
**3.1  Weather conditions**
Comparing the air temperature of the study period to the long-term average (8.7 °C), the first period
(2011/12) was warmer (9.6 °C), the second period (2012/13) was colder (8.1 °C) and the third period
(2012/14) was warmer again (9.8 °C). The precipitation sums of the first two study periods (1012 mm
in 2011/12 and 971 mm in 2012/13) were higher than the long-term annual precipitation sum (861 mm),
whereas precipitation was lower in the third study period (821 mm).
Considerable differences between the three periods are consisting in days with mean temperatures
below 0 °C (Fig. 1). While in the first and third winter only one period with 20 and 11 frost days,
respectively, occurred, several freeze/thaw events and in total 58 days with mean temperatures below
the freezing point appeared in the second winter. Therefore, in 2013 the vegetation period started about
one month later than in 2012 and 2014. High precipitation events took place in August 2011, leading to
a precipitation sum of 236 % the long-term average for this month (Fig. 1). Above-average precipitation
also occurred in July 2012 (183 %), whereas in summer 2013 only 41 and 58 % of long-term average
precipitation were registered in July and August, respectively.



## 3.2 Groundwater levels
Groundwater levels (GWLs) during the study period showed high variability between sites and years
(Fig. 2). Highest fluctuations were recorded on sites GM and AR with the same minima and maxima of
-88 and 2 cm, respectively. Variability was lower at sites UG and GW with minima of -56 (UG) and -65
cm (GW) and maxima of 8 (UG) and 2 cm (GW) for the 3-year period. Also short-term fluctuations
with GWLs close to the soil surface and deep water levels within a few days or weeks were more
distinct at sites GM and AR. In summers 2011 and 2012, all sites showed high GWLs close to the
surface and even periods of inundation at site UG, whereas in summer 2013 GWLs were considerably
lower (Fig. 2).
## 3.3 GHG fluxes
### 3.3.1 Methane ($CH_4$)
Daily methane fluxes were highest at site UG and low at the agricultural sites (Fig. 3). While the
intensively drained sites GM and AR showed negligible $CH_4$ exchange, $CH_4$ fluxes were on a higher
level at site GW with one distinct emission peak in April 2013. $CH_4$ emissions from site UG showed
high spatial and temporal variability. Emissions increased for the first time in August 2011, followed by
a continuous release of $CH_4$ until July 2013. Highest emission peaks were recorded in summer 2012 and
after that high releases occurred in autumn 2012 and spring 2013. Remarkably, the $CH_4$ flux pattern at
site UG changed substantially in July 2013 as emissions ceased and did not rise again until the end of
the study period in spring 2014.
### 3.3.2 Nitrous oxide ($N_2O$)
$N_2O$ fluxes during the 3-year period showed no distinct regularity at the unutilized site (UG), whereas
the agricultural sites showed seasonal flux patterns with several emission peaks during spring, mainly
occurring after N fertilization (Fig. 4). While emissions at site UG peak in May 2013, the highest $N_2O$
releases from site GW were observed in April 2012. Similar but more frequent emission peaks were
recorded at site GM in April and May 2012 and 2013 and further distinct $N_2O$ releases from that site
were observed in autumn and winter 2013. The most pronounced seasonality of $N_2O$ emissions was
determined at the arable site (AR) with high releases at the beginning of each study period. Thereby, the
emissions in May 2013 clearly exceeded those of the preceding two years.



### 3.3.3 Carbon dioxide ($CO_2$)

The carbon dioxide exchange of the study sites was characterized by seasonal patterns of gross primary production (GPP) and ecosystem respiration ($R_{ECO}$) with high exchange rates during the vegetation period and smaller fluxes between October and April (Fig. 5). Maximum $CO_2$ uptake rates were -176, -188, -228 and -320 kg $CO_2$-C ha$^{-1}$ d$^{-1}$ for sites UG, GW, GM and AR, respectively (SE 9, 7, 17, 11). While this maximum C fixation took place in July 2013 at site UG, the two utilized grassland sites showed highest productivity in May 2012 before the first cutting. At site AR, maximum $CO_2$ uptake was modelled for the spring barley in June 2012. After the harvest of barley in August 2012, weeds remained that were eliminated by pesticides and mulched in September, so no $CO_2$ uptake could occur until emergence of newly seeded plants in May 2013. Maximum modelled $CO_2$ releases by $R_{ECO}$ from sites UG, GW, GM and AR were 156 (August 2012), 231 (May 2012), 216 (August 2012) and 259 kg $CO_2$-C ha$^{-1}$ d$^{-1}$ (June 2012), respectively (SE 16, 6, 2, 11). Depending on the extent of daily GPP and $R_{ECO}$ fluxes, the sites can act as net source or sink for $CO_2$. In total of two years (730 days), sites UG, GW, GM and AR acted as a $CO_2$ sink on 182, 156, 102 and 115 days, whereas they showed a net $CO_2$ release on 548, 574, 628 and 615 days, respectively (Fig. 5).

### 3.4 GHG balances, NECB and GWP

### 3.4.1 $CH_4$ and $N_2O$ balances

Over the three-year study period, mean annual $CH_4$ emissions were 55.1, 13.5, 0.9 and 1.8 kg $CH_4$-C ha$^{-1}$ a$^{-1}$ for sites UG, GW, GM and AR, respectively (SE 17.2, 4.0, 0.5, 0.7). Highest annual $CH_4$ release occurred at site UG in the second year, while minimum budgets were determined for sites GM and AR in the third year (Table 2). However, due to the low fluxes at sites GM and AR, cumulated annual $CH_4$ emissions were not significantly different from zero ($p > 0.05$). Sites GW and UG represented sources for $CH_4$ with significantly higher releases at site UG that also showed the highest variation in annual $CH_4$ budgets (Fig. 6a). Mean annual $N_2O$ balances of the four sites increased in the order UG, GW, GM and AR, accounting for 3.4, 6.5, 14.4 and 18.9 kg $N_2O$-N ha$^{-1}$ a$^{-1}$, respectively (SE 0.6, 0.9, 2.0, 1.1). Highest annual $N_2O$ emissions were recorded at site AR in the third year, whereas site UG released minimum amounts of $N_2O$ in the second year (Table 2). The high budgets of sites GM and AR showed high variation and thus, did not differ significantly ($p = 0.18$) (Fig. 6b).



### 1   3.4.2  CO$_2$ balances and NECB

For the two years of CO$_2$ exchange measurement, mean annual NEE was 2.8, 8.0, 11.7 and 10.1 Mg
CO$_2$-C ha$^{-1}$ a$^{-1}$ for sites UG, GW, GM and AR, respectively (SE 2.5, 0.7, 1.2, 1.9) (Fig. 7a). Thus, all
sites showed higher annual R$_{ECO}$ than GPP sums, with highest R$_{ECO}$ at site AR and lowest R$_{ECO}$ at site
UG, both for the period 2013/14 (Table 2). Highest annual GPP was determined at site AR for 2013/14,
whereas site GM showed lowest GPP during the same period. As for R$_{ECO}$ and GPP, both highest and
lowest NEE occurred in 2013/14 at sites GM and UG, respectively (Table 2). As indicated by NECB,
all sites were net C sources during the study period with mean annual losses of 2.8, 10.6, 15.7 and 15.0
Mg C ha$^{-1}$ a$^{-1}$ at sites UG, GW, GM and AR, respectively (SE 2.6, 1.1, 1.4, 2.4) (Fig. 7b). Consistent
with NEE, a higher range of NECB was assessed for the period 2013/14 with lowest C losses at site UG
and highest losses at site AR (Table 2). The NEE and NECB of sites GW and AR did not differ
significantly (Figs. 7a and b). However, mean NECB of site AR tended to be higher compared to site
GW with p = 0.07.

### 14   3.4.3  GWP

The GWP combines the CO$_2$-C-eqs of NEE, CH$_4$ and N$_2$O emissions, as well as the anthropogenic C
balances from slurry applications and biomass removals. For the study periods 2012/13 and 2013/14,
mean annual GWP was 3.8, 11.7, 17.7 and 17.3 Mg CO$_2$-C-eq ha$^{-1}$ a$^{-1}$ for sites UG, GW, GM and AR,
respectively (SE 2.8, 1.2, 1.8, 2.6) (Fig. 7c). The lowest (site UG) as well as the highest GWPs (site
AR) were observed for 2013/14 (Table 2). NEE dominated GWP at all sites with mean shares ranging
from 59 % at site AR to 72 % at site UG. However, as no biomass removal occurred on site UG, this
site also showed the highest shares of CH$_4$ and N$_2$O, with each gas accounting for 14 % of the GWP on
average of the two years. The GWPs of the agricultural sites were considerably influenced by the C
balances of slurry inputs and harvested biomass, which accounted for 21, 23 and 27 % at sites GW, GM
and AR, respectively.

### 25   3.5   Crop yields and yield related GHG emissions

### 26   3.5.1  Biomass, carbon, nitrogen and energy yields

For the grassland sites, all yield parameters were higher in 2012 than in 2013 (Table 3). While this
reduction was significant for site GM, site GW showed no significant differences between years. At the
arable site, significantly higher yields were obtained by spring wheat with undersown grass in 2013
compared to spring barley in 2012. Site GM revealed significantly higher yields than site AR in 2012,
while site GW did not differ to any other site in that year, except for N yield. In 2013, yields of sites





GM and AR showed no significant differences, while site GW had significantly lower yields than the
other two sites, except for the N yield of site AR and the NEL yield of site GM. On average, site GM
showed the highest yields, while lowest yields were observed on site GW, except for N yield, which
was lowest on site AR. However, only N yield of sites GM and AR differed significantly.
**3.5.2  Yield related GHG emissions**
The annual GWP (Table 2) was related to the annual energy yields (Table 3) of the three agricultural
study sites. While these yield related GHG emissions increased for site GM in the second year, they
decreased for sites GW and AR (Table 4). On average of the two year study period, site GM did not
differ significantly to the other sites, whereas site GW showed significantly lower yield related
emissions than site AR.
**4    Discussion**
**4.1    CH$_4$ fluxes and balances**
Sites GM and AR showed negligible CH$_4$ fluxes and annual CH$_4$ budgets were not significantly
different from zero. This is in accordance with other observations on intensively used peat soils that
report low CH$_4$ emissions or even net uptake of CH$_4$ (Flessa et al., 1998; Maljanen et al., 2003a, 2004;
Schäfer et al., 2012). The water table is the main controlling factor for CH$_4$ emissions from peat soils,
particularly in absence of aerenchymus shunt species. A drainage depth of $20 - 30$ cm is regarded as
sufficient to inhibit the diffusion of high amounts of CH$_4$ into the atmosphere as CH$_4$ produced in the
anoxic zone is oxidized by methanotrophs in the unsaturated zone (Couwenberg, 2009a; Schäfer et al.,
2012). Accordingly, the low CH$_4$ fluxes at sites GM and AR can be explained by the high drainage
intensity. However, a high GWL close to or above the soil surface did not enhance CH$_4$ production and
emission at these sites (Fig. 8). A multiple linear regression model showed significant relations between
log-transformed daily CH$_4$ fluxes and site ($p < 0.001$), GWL ($p < 0.001$) and soil temperature at 5 cm
depth ($p < 0.01$). Therefore, reactions on alterations of GWL and soil temperature differed between
sites, probably as a consequence of long-term adaptation of methanogenic and methanotrophic
communities to drainage intensity (van den Pol-van Dasselaar et al., 1997; Yrjälä et al., 2011). At site
GW, CH$_4$ production potential was higher compared to sites GM and AR, leading to considerable CH$_4$
releases, especially when GWL and soil temperature were high, as for example in summer 2012 (Fig.

30  3).



Conspicuous $CH_4$ peaks were detected at site UG in 2012 (Figs. 3 and 8) that were associated with high
GWLs due to heavy rain fall in July and high soil temperatures due to a heat wave in late July and
August (Figs. 1 and 2). These conditions likely favored a rapid expansion of the methanogenic
community, more pronounced than in summer 2011 when GWLs were similarly high but temperatures
were lower. Nykänen et al. (1998) reported that peat temperature controls $CH_4$ dynamics at high water
tables, whereas the correlation is poor at low water tables. This is confirmed by the situation at site UG
in summer 2013 when $CH_4$ emissions ceased as a consequence of low precipitation and water level
drawdown in July and August, although soil temperatures were high. The subsequent GWL rise in
autumn had no effect on $CH_4$ emissions, which remained low until the end of the study period. A
possible explanation is that the methanogenic community was impaired by oxidative stress in summer
(Görres et al., 2013) and did not recover due to low soil temperature when GWL rose (Bubier & Moore,
1994). Knorr et al. (2008) reported that $CH_4$ production in a fen soil was retarded by experimental
drought for up to several weeks after rewetting. Estop-Aragonés & Blodau (2012) observed a longer
time lag until $CH_4$ production recovered after rewetting for more intense and longer dried fen peat but
warmer conditions favored the recovery. Furthermore, the dry soil conditions in summer 2013 could
have increased the methanotrophic community, leading to a $CH_4$ consumption potential in the
subsequent months exceeding the production potential as methanotrophic bacteria react less sensitively
to temperature changes than methanogenic bacteria (Dunfield et al., 1993). This is supported by the
results of this study as the overall highest daily $CH_4$ uptakes were measured at site UG in summer and
autumn 2013.
Annual $CH_4$ balances of the study sites are comparable to those recently reported for temperate
European peat soils (Schäfer et al., 2012; Beetz et al., 2013; Leiber-Sauheitl et al., 2014). Annual
balances were significantly related to site and mean annual GWL (both with p < 0.001). Confirming the
general understanding of $CH_4$ emission patterns (Couwenberg, 2009a), no significant $CH_4$ releases were
observed for mean GWLs below -25 cm. At mean GWLs above -10 cm, $CH_4$ emissions were highly
variable, with a minimum release of 28 and a maximum of 430 kg $CH_4$-C ha$^{-1}$ a$^{-1}$ (Fig. 10a), which is
typical for the high spatial variability of $CH_4$ fluxes (Waddington & Roulet, 1996; van den Pol-van
Dasselaar et al., 1999). The low contribution of $CH_4$ emissions to the GWP of the three agricultural sites
(Table 2) illustrates the minor importance of $CH_4$ in terms of GHG mitigation on utilized peat soils.
However, Hahn-Schöfl et al. (2011) showed that degraded fen grasslands can emit huge amounts of
$CH_4$ as a consequence of flooding when easily degradable fresh plant material is present. Therefore,
inundation of sites with highly productive, energy rich grasses such as perennial ryegrass (*Lolium*
*perenne*) bears the risk of enhanced $CH_4$ emissions, especially during summer. This should be
particularly considered for site GW, where a significant $CH_4$ production potential could be observed.



## 2  4.2  $N_2O$ fluxes and balances

$N_2O$ emissions measured at the study sites were of similar magnitude as observed for other agricultural
fen soils, for example in South Germany (Flessa et al., 1998), the Netherlands (van Beek et al., 2010;
2011) or Denmark (Petersen et al., 2012) and conform to the range of $N_2O$ hotspots on European
organic soils given by Leppelt et al. (2014). The $N_2O$ release from site UG represents the emissions
without agricultural utilization in the study area. These were higher than reported for natural peatlands
(Leppelt et al., 2014), which might be a result of GWL fluctuations (Figs. 2 and 10b), as background
$N_2O$ emissions strongly depend on drainage intensity (van Beek et al., 2011). A multiple linear
regression model for log-transformed daily $N_2O$ fluxes gave significant effects of site and the amount of
nitrate in 0 – 20 cm soil depth (both with $p < 0.001$) with highest fluxes measured at high soil nitrate.
Soil nitrate contents are enhanced by mineral fertilizer inputs on the one hand and mineralization and
nitrification of organic N in soil organic matter (SOM) or organic fertilizers on the other hand. Several
$N_2O$ emission peaks at the three agricultural study sites occurred subsequent to mineral fertilizer or
slurry application, especially at site AR and in spring 2012 at all three sites (Fig. 4). High soil nitrate,
exceeding the current N uptake capacity of vegetation can cause increased $N_2O$ production through
denitrification, thus N fertilization often leads to enhanced $N_2O$ emissions for several days to weeks
(Velthof & Oenema, 1995b; Bouwman et al., 2002; Grant et al., 2006). In addition, a nitrate surplus in
soil promotes incomplete denitrification and increasing $N_2O/N_2$ product ratios with the associated risk
of $N_2O$ emissions (Firestone et al., 1980; Farquharson & Baldock, 2008; Senbayram et al., 2012). At
site AR, strong $N_2O$ emission peaks occurred after fertilization in spring when vegetation was missing
or seeded plants were emerging (Fig. 4).
Therefore, instead of relating annual $N_2O$ emissions to annual N balances, short-term N balances for
about two week intervals were calculated for site AR and the vegetation periods 2012 and 2013 and
related to the $N_2O$ balances of the same period. This was conducted by considering the N input by
fertilizers as well as the N uptake by plants (Fig. 9). During the first weeks after fertilizer application, N
surpluses of up to 99 kg ha$^{-1}$ occurred, leading to extremely high short-term $N_2O$ releases in some cases.
The increasing N uptake in the subsequent periods was characterized by N balances ranging from -48 to
12 kg N ha$^{-1}$ without significant $N_2O$ emissions. These findings confirm to a meta-analysis of van
Groenigen et al. (2010), who found no differences in $N_2O$ emissions for negative or slightly positive N
balances, but significantly increasing emissions for a surplus of 90 kg N ha$^{-1}$. During the period
2012/13, 73 % of $N_2O$ emissions at site AR occurred in April and May, while for the period 2013/14, 90
% of the total annual $N_2O$ budget was emitted in May. Therefore, it can be concluded that in



combination with tilling, which might increase the availability of easily decomposable organic C for
denitrifiers (Nykänen et al., 1995), fertilization of peat soils during periods with lacking N uptake
capacity, bears the risk of substantial $N_2O$ emissions (Maljanen et al., 2003b; Regina et al., 2004).
After a second smaller fertilization peak at site AR in June 2013, $N_2O$ emissions were reduced to zero
or even small uptakes of $N_2O$ were detected (Fig. 4), which can be explained by increased vegetation
productivity. The growing plants act as competitor for nitrate to the denitrifiers, leading to complete
denitrification as nitrate availability is strongly decreased. This was described for pristine (Roobroeck et
al., 2009) or restored peatlands (Silvan et al., 2005) were N availability is usually limited (Martikainen
et al., 1993). Our results suggest that on sites with very high $N_2O$ production potential, emissions can be
eliminated by a continuous coverage of highly productive plants and prevention of fertilization when N
uptake is limited. $N_2O$ uptake into soils is often linked to low mineral N and high moisture contents
(Chapuis-Lardy et al., 2007). However, the small but continuous $N_2O$ uptakes at site AR, beginning in
June 2013, were probably attributed to a high denitrification potential, stimulated by the excess of
nitrate during May, and a shift to $N_2O$ consumption by denitrifiers when nitrate competition by plant
roots increased (Roobroeck et al., 2009).
On average, $N_2O$-N emissions from the agricultural study sites accounted for 2.2, 5.9 and 13.2 % of
applied N for sites GW, GM and AR, respectively. The values for sites GW and GM fit well with those
presented by van Beek et al. (2010) for grazed grasslands on organic soil in the Netherlands with
comparable GWLs. Therefore, our results support the findings of van Beek et al. (2010), who argued
that mean annual GWL should be used in addition to N input for estimating $N_2O$ emissions from
organic soils, as the ratio of $N_2O$ emissions to N input increases with decreasing GWLs. However, our
results illustrate that the type of management should be considered as well, as arable cropping can
induce a disproportional increase of $N_2O$ emissions related to N input.
Drained organic soils are known to emit significant shares of their annual $N_2O$ budget during the winter
period (Priemé & Christensen, 2001; Maljanen et al., 2003b), increasing with the number of freezing
and thawing cycles (Regina et al., 2004). Thereby, $N_2O$ emissions are enhanced during freezing as well
as thawing, since both processes release C into the soil, which is rapidly utilized by heterotrophic
denitrifiers (Koponen et al., 2006). In contrast, emission peaks during winter were observed in the first
and third year when only one period with negative temperatures occurred, but not in the second year,
when more freezing and thawing cycles appeared (Figs. 1 and 4). The reason might be the deeper frozen
soils in the first and third winters, as no snow cover was present, inducing higher C releases.
Annual $N_2O$ emissions were significantly related to mean annual GWL (Fig. 10b), which might be
explained by increasing amounts of nitrate in top soil with increasing drainage intensity (Fig. 11a). As



the differences in soil nitrate could not be attributed to different N fertilization intensities (Table 1), the
GWL seemed to control nitrification processes. Koops et al. (1997) emphasized that nitrification is an
important process for $N_2O$ losses from peat soils, while Dowrick et al. (1999) stated that denitrification
is the main source for $N_2O$ emissions from drained organic soils as the nitrate produced from peat
mineralization is reduced in small-scale anaerobic porosity. However, both nitrification and
denitrification processes likely contributed to $N_2O$ emissions as sites GM and AR show strong
fluctuations in GWL (Fig. 2), which generally lead to a pronounced cycling of both processes and thus
enhanced $N_2O$ release (Goldberg et al., 2010; Jørgensen & Elberling, 2012).
**4.3   CO$_2$ exchange and NECB**
All four study sites were net C sources during the two years of $CO_2$ measurements (Table 2 and Fig. 7).
Compared to IPCC (2014) emission factors for temperate organic soils, the sites showed NEE values
above the given range for their respective land use categories. While the NEE of site AR was 9.0 Mg
$CO_2$-C ha$^{-1}$ a$^{-1}$ in 2012/13, which is within the 95 % confidence interval of 6.5 – 9.4 Mg $CO_2$-C ha$^{-1}$ a$^{-1}$
given by IPCC (2014) for drained temperate croplands, it was above that range in 2013/14 (11.2 Mg
$CO_2$-C ha$^{-1}$ a$^{-1}$). The NEE of sites GM and GW exceeded the intervals for nutrient-rich temperate
grasslands that are deep-drained (5.0 – 7.3 Mg $CO_2$-C ha$^{-1}$ a$^{-1}$) or shallow-drained (1.8 – 5.4 Mg $CO_2$-C
ha$^{-1}$ a$^{-1}$) in both years (Table 2). If the NECB is considered, the C losses of the agricultural sites were
even higher, thus exceeding the upper values of IPCC emission factors for the respective land use
categories by a factor of 2.0, 2.2 and 1.6 for sites GW, GM and AR, respectively. Moreover, the C loss
from site UG clearly exceeded the average IPCC emission factor for rewetted and nutrient-rich
temperate organic soils of 0.5 Mg $CO_2$-C ha$^{-1}$ a$^{-1}$ in both years.
Recently published results for utilized organic soils in the same climatic region as the study area of this
observation showed net C losses of 4.3 – 8.2 Mg $CO_2$-C ha$^{-1}$ a$^{-1}$ for an intensively managed peat bog
grassland in Germany (Beetz et al., 2013), 3.3 – 8.6 Mg $CO_2$-C ha$^{-1}$ a$^{-1}$ for extensively managed
grasslands on *histic Gleysol* in Germany (Leiber-Sauheitl et al., 2014) and 6.9 – 16.7 Mg $CO_2$-C ha$^{-1}$ a$^{-1}$
for grassland and arable cropping on bog and fen soils in Denmark (Elsgaard et al., 2012). The highest
value of 16.7 Mg $CO_2$-C ha$^{-1}$ a$^{-1}$ represented a rotational grassland on fen soil, thus a comparable system
to site AR in 2013/14, which showed a similar NECB of 17.7 Mg $CO_2$-C ha$^{-1}$ a$^{-1}$. However, the NEE of
the Danish site was even higher (13.6 Mg $CO_2$-C ha$^{-1}$ a$^{-1}$) than at site AR (11.2 Mg $CO_2$-C ha$^{-1}$ a$^{-1}$),
indicating that C removal by harvest from site AR was comparatively high. The permanent grassland
sites studied by Elsgaard et al. (2012) showed C losses between 6.9 and 10.4 Mg $CO_2$-C ha$^{-1}$ a$^{-1}$. In
conclusion, C losses of sites UG, GW and AR were at the upper end of literature values, while the



NECB of site GM clearly exceeded the given ranges. The comparatively high C losses of the study sites
highlight the functioning of the study region as a considerable C source, underlining the need for
mitigation strategies.
Seasonal variability of NEE on agricultural grasslands cannot only be explained by environmental
parameters as their influence is often superposed by management activities like grassland cuttings
(Wohlfahrt et al., 2008b). Land use intensity affects the NEE of ecosystems, as the frequency of
biomass removals influences respiration processes as well as photosynthesis (Soussana et al., 2007).
Generally, it is assumed that NEE increases with the number of cuttings, since GPP is reduced to almost
zero for several days after harvest, while $R_{ECO}$ can remain high, depending on the extent of soil
respiration (Schmitt et al., 2010). At the studied grassland sites, $R_{ECO}$ was often reduced by cutting
events but not in the same degree as GPP, leading to sharp increases of NEE after harvest (Fig. 5). The
effect of an increased number of grassland cuttings was especially pronounced at site GM, where four
cuttings were conducted in the second year, compared to three cuttings in the first year. Thereby, $R_{ECO}$
was reduced to a greater extent than GPP, leading to a slightly increased NEE. However, at site GW the
effect was different when the number of cuttings increased from two in the first to three in the second
year. Here, a smaller $R_{ECO}$ but slightly increased GPP resulted in a lower NEE in the second year. The
same effect was visible for GPP when comparing sites GM and GW for a given year (Table 2). These
results suggest that changing grassland management from two to three cuttings per year did not reduce
total annual photosynthetic activity, while GPP could be diminished by four cuttings. However,
irrespective of total number of grassland harvests, the first cuts were performed in May, the common
time for intensively managed grasslands as the average growth rate is at its maximum (Parsons &
Chapman, 2000). Before the first cut, the NEE of grasslands is mainly controlled by GPP (Wohlfahrt et
al., 2008a). Shifting the first cut to June or July would therefore increase the total productivity of first
growth period and extend the phase of net $CO_2$ uptake. However, this is hardly compatible to intensive
grassland management depending on profitability (McInerney, 2000) as forage quality would be too
low. After a grassland cut it took several weeks until the sites showed net $CO_2$ uptake again, often
closely followed by the next cutting (Fig. 5). Therefore, the cutting regime strongly controlled the NEE
of the agricultural grassland sites.
Unutilized peatland ecosystems can either be sources or sinks for $CO_2$, depending on variables like
trophic status, peat temperature, water table (Bubier et al., 1998) or vegetation composition (Leppälä et
al., 2011). As the difference between uptake (GPP) and release of $CO_2$ ($R_{ECO}$) is generally small,
marginal changes of these parameters can invert the NEE of a peatland between different years (Bubier
et al., 1999; Griffis et al., 2000; Arneth et al., 2002). At site UG, maximum daily GPP was observed in
July, followed by a decrease in August, while $R_{ECO}$ reached its maximum a few weeks later then



declined to a lesser extent. This was typical as the annual course of $R_{ECO}$ is usually shifted by about one
month compared to GPP (Lloyd & Taylor, 1994). Consequently, daily $CO_2$ uptake reaches its maximum
in spring or early summer and a net release of $CO_2$ starts in late summer when vegetation becomes
senescent and $R_{ECO}$ exceeds GPP (Bellisario et al., 1998; Parsons & Chapman, 2000). A late cutting of
vegetation could delay senescence and prolong the period of plant growth at site UG, which might
reduce NEE. However, Beetz et al. (2013) observed that a single cutting event shifted a rewetted and
extensively used peat bog grassland from a $CO_2$ sink to a small source as annual GPP was reduced by
more than annual $R_{ECO}$. This cutting was, however, conducted at the end of vegetation period and GPP
did not rise again. The optimum time for a one-cutting grassland system in terms of maximizing GPP by
avoiding early senescence might be in late July or early August to take advantage of both a highly
productive primary growth and regrowth period. In addition, this was usually the period of lowest
groundwater levels (Fig. 2), ensuring the viability of a grassland cutting as the limit for trafficability on
fen soils is a GWL around -30 cm (Blankenburg et al., 2001). However, a potentially smaller NEE of a
one-cut system might be offset by an increase in NECB due to biomass removal.
At site AR, the change of management with undersown grass in 2013 greatly influenced the courses and
annual sums of GPP and $R_{ECO}$ (Table 2 and Fig. 5). Both increased in the second year due to a
continuous plant cover but with a larger increase of $R_{ECO}$, resulting in a higher NEE. As the C export by
harvest also increased considerably (Table 3), the change of NECB was even greater than for NEE. In
2012, no plants remained on the site after pesticide application and mulching in September, eliminating
GPP and autotrophic respiration ($R_a$). Due to a wet summer, harvest was conducted late and in spite of a
high GWL, which induced soil compaction. In combination with the lack of water removal by plants,
this led to inundation during autumn and winter. As a consequence, soil respiration was low during
winter 2012/13 (Fig. 5). In contrast, $R_{ECO}$ and GPP fluxes were higher in winter 2013/14 and
considerably increased at the end of the study period due to highly productive new established grass, a
lower GWL (Fig. 2) and higher temperatures (Fig. 1).
Several studies observed increasing $CO_2$ emissions from peatland ecosystems with increasing drainage
intensity (e.g. Moore & Knowles, 1989; Bubier et al., 1998; Drösler, 2005; Dinsmore et al., 2009).
Since the variability of NEE for an individual agricultural site strongly depends on management
(Wohlfahrt et al., 2008b) as described above, inter-site comparison is necessary to illustrate the effect of
water level on NEE. On average of the four study sites and both years, NEE significantly increased by
about 220 kg $CO_2$-C ha$^{-1}$ a$^{-1}$ per cm lowering of mean annual GWL (Fig. 12a). Moreover, our results
suggest that arable cropping of peatlands did not lead to higher $CO_2$ emissions per se, confirming recent
observations from peatland sites in Germany (Drösler et al., 2013) and Denmark (Elsgaard et al., 2012).
Despite a lower mean annual GWL on site AR (Table 1), NEE and NECB of sites AR and GM did not





differ significantly (Fig. 7). This can be explained by a lower $R_{ECO}$ due to missing vegetation cover and
water logging after harvest at site AR in the first year and a very high GPP due to undersown grass in
the second year. Furthermore, Estop-Aragonés et al. (2012) argue that in compacted peat soils with high
bulk densities and ash contents, oxygen penetration is reduced compared to less compacted soils,
resulting in lower air filled porosity and soil respiration. Due to the higher peat degradation of site AR
(Table 1), this could partly explain the similar NEE of sites AR and GM.
While Aurela et al. (2007) reported that a drought period in a Finnish sedge fen increased $R_{ECO}$ and thus
NEE, Leppälä et al. (2011) concluded that the difference in NEE between wet and dry years for natural
peatlands in Finland resulted from alterations of GPP rather than $R_{ECO}$. For the dryer second year of our
observations, $R_{ECO}$ of site UG was lower than in the first year, while GPP decreased only marginally
(Table 2). However, comparing only July and August, the period with greatest difference in GWLs
between the years (-9.2 cm in 2012 and -36.6 cm in 2013; Fig. 2), $R_{ECO}$ was almost the same (6.9 and
6.8 Mg $CO_2$-C ha$^{-1}$ in 2012 and 2013, respectively), which is in line with results presented by
Parmentier et al. (2009) and Muhr et al. (2011). GPP slightly increased in the drier year (-7.2 and -7.6
Mg $CO_2$-C ha$^{-1}$ in 2012 and 2013, respectively). As main reason for differences in NEE between the
two years, the weather conditions in spring could be identified. In 2012, the growing season, calculated
by the method of Janssens (2010), started on 20 March, while it was delayed by more than one month in
2013 to 23 April. As a result of different weather conditions, cumulated $R_{ECO}$ for April and May was 4.1
Mg in 2012 and only 2.2 Mg $CO_2$-C ha$^{-1}$ in 2013. Besides, GPP was -3.2 Mg for April and May 2012
and -2.3 Mg $CO_2$-C ha$^{-1}$ in 2013. These differences cannot be explained by mean GWL for the two
months (-4.9 cm in 2012 and -8.4 cm in 2013) as the different weather conditions were the dominating
parameter. Thus, respiration processes were stimulated more than plant productivity by the earlier start
of growing season, indicating that shorter winter periods potentially increase the risk of higher C losses
from peatland ecosystems. Griffis et al. (2000) studied the NEE of a subarctic fen and concluded that
the phenological stage of vegetation relative to the climatic conditions is important for interannual
variability of NEE. In conclusion, the mean GWL of single years cannot be solely used to predict the
variability of NEE at the same site or between sites with different management as climatic and
management effects can be of dominating importance.

## 4.4  Global warming potential

The global warming potential (GWP) of the four study sites increased in the same order as NEE and
NECB. However, the difference between sites GW and AR was significant for GWP whereas it was not
significant for NEE and NECB (Fig. 7). This can be explained by significantly higher $N_2O$ emissions at




site AR (Fig. 6b). NEE mainly controlled the GHG balances, accounting for 72, 69, 66 and 59 % of the
GWP on sites UG, GW, GM and AR, respectively. In addition, the balances of C export via harvest and
C import via slurry contributed considerably to the GWP of the agricultural sites, accounting for 21, 23
and 27 % for sites GW, GM and AR, respectively, indicating a higher share of anthropogenic C fluxes
with higher land use intensity. Compared to other observations or reviews of peatland GHG emissions
in northern or temperate Europe, the GWP of the study sites was at the upper end of presented emission
factors (Nykänen et al., 1995; Langeveld et al., 1997; Kasimir-Klemedtsson et al., 1997; Alm et al.,
2007a; Oleszczuk et al., 2008; Maljanen et al., 2010; Drösler et al., 2013).
Site UG showed a significantly lower GWP compared to the agricultural sites, supporting the
assumption that rewetting of drained organic soils reduces their climatic footprint (Höper et al., 2008;
Beetz et al., 2013). The lower GWP of site UG was a result of missing C losses through harvest and
reduced $CO_2$ and $N_2O$ emissions that could mainly be attributed to the high GWLs (Fig. 12b),
outweighing the higher $CH_4$ release (Fig. 6a). A linear regression for all four sites and both years gave a
significant increase of GWP for about 410 kg $CO_2$-C-eq $ha^{-1}$ $a^{-1}$ per cm lowering of mean annual GWL
(Fig. 12b). The higher slope compared to NEE (Fig. 12a) was a result of $N_2O$ emissions, significantly
increasing with drainage intensity as well (Fig. 10b). However, as $CH_4$ emissions tended to increase
exponentially when water levels were close to the soil surface (Fig. 10a), the slope might decline or
even invert for a mean annual GWL around or above 0 (Augustin & Joosten, 2007). Therefore, the
intercept of ~2 Mg $CO_2$-C-eq $ha^{-1}$ $a^{-1}$ should not be over-interpreted. A mean annual GWL of about 10
cm below the soil surface is often referred to as an optimum scenario for mitigating GHG emissions
from peatlands, as $CO_2$ emissions are greatly reduced or even negative (i.e. $CO_2$ uptake) and $CH_4$ fluxes
are hampered by the small oxic horizon (e.g. Couwenberg et al., 2011). However, this is not only
controlled by mean annual GWL, but equally by groundwater fluctuations (Dinsmore et al., 2009).
Thus, the relatively high GWP of site UG (3.8 Mg $CO_2$-C-eq $ha^{-1}$ $a^{-1}$) in spite of a high mean annual
GWL (Table 1) suggests that a further increase and stabilization of water levels might be necessary to
reduce the climatic impact of that site. The lack of natural, peat forming mire vegetation (Table 1)
supports this assumption as the GWP of natural or rewetted reed and sedge fens is assumed to be around
1 Mg $CO_2$-C-eq $ha^{-1}$ $a^{-1}$ (Couwenberg et al., 2011; Drösler et al., 2013).

## 30    4.5   Yield related emissions

To assess the climatic footprint of the agricultural study sites, their function in terms of forage and milk
production has to be considered in addition to area related GHG emissions. On average of two years,
site GW represented the most climate-efficient forage production system of the three sites, whereas site



AR caused the greatest GHG emissions relative to energy yield (Table 4). Observations of greenhouse
gas emissions from arable forage cropping systems at two sites on mineral soil in northern Germany
resulted in yield related emissions between -18 and 32.5 kg $CO_2$-C-eq (GJ NEL)$^{-1}$, including all
emissions during crop production, transport and storage (Herrmann et al., 2014). Hence, the field based
emissions at the study sites presented here, demonstrate that forage produced on fen soils is burdened
with many times higher GHG emissions compared to forage from mineral soils of the same region.
The high yield related emissions of site AR were mainly attributed to the low energy yield of barley in
the first year, resulting from wet conditions in summer and thus a delayed harvest with low quality for
milk production. In addition, the site was only partially harvested due to high soil moisture, thus, the
'true yield' per ha was even lower than given in Table 3. Moreover, the maize in 2011 could not be
harvested at all due to above-average precipitation in August and September (Fig. 1). Therefore, arable
forage production on fen soils of the study area is associated with a high uncertainty of yield in wet
years, which, considering the high GHG emissions, makes it an inappropriate type of management from
both an economic and environmental point of view. Underlining this conclusion, the management of
site AR was changed in 2013 with undersown grass, increasing the certainty of yield as the time of
harvest became more flexible. However, despite a high yield in 2013, yield related emissions remained
higher compared to site GW as a result of a very high GWP (Table 2).
Comparing the two grassland sites, the four-cut system of site GM in 2013 showed the highest and the
three-cut system of site GW in 2013 induced lowest yield related GHG emissions. In addition, the two-
cut system of site GW in 2012 had higher yield related emissions than the three-cut systems of both
sites. Therefore, a three-cut grassland in combination with a preferably high GWL represented the most
climate-efficient management system at the studied fen soils. On average of both years, the energy yield
of site GW was 19 % lower compared to site GM, while the GWP was 34 % lower. This difference was
only significant for GWP. Thus, the effect of a raised water level can be assumed to be greater for GHG
emission reduction than for yield reduction. This is in line with results of Renger et al. (2002), who
reported that for a mean GWL of -30 cm, 90 % of optimum plant output can be reached, while GHG
emissions can be reduced for 40 – 50 % of maximum emissions. These values were obtained by a water
regime model and represent an optimum scenario, indicating that further potential exists to improve the
climate efficiency of forage production on site GW. Reasons for reduced productivity on poorly drained
soils could include the loss of sown species in favor of undesirable species with increasing sward age
(Hopkins & Green, 1979) and a lower soil warming in spring due to high soil moisture, resulting in
delayed plant growth (Tyson et al., 1992). The first aspect was evident in increasing shares of creeping
bentgrass (*Agrostis stolonifera*) and water foxtail (*Alopecurus geniculatus*) at site GW, indicating the
need for occasional resowing of productive species like perennial ryegrass (*Lolium perenne*).





None of the conventional management options can be regarded as sustainable in terms of peat
conservation as each type of utilization associated with peatland drainage led to peat mineralization
(Joosten & Clarke, 2002; Renger et al., 2002). The ongoing subsidence due to peat loss might chance
the utilization structure in future as sites become wetter and some areas might need to be extensified or
abandoned, opening potentials for GHG mitigation. This was recently evident at site GW, where only
two cuts could be realized in the wet years 2011 and 2012.
## 5   Conclusions
Long-term drainage intensity was the most important controlling factor for GHG emissions from the
studied fen soils. NEE dominated the GHG balances of all sites and as assumed, considerable
differences in GHG fluxes and balances were observed among the sites. After 20 years of rewetting (site
UG), emissions of $CO_2$ and $N_2O$ were significantly lower while significantly higher amounts of $CH_4$
were emitted compared to the agricultural sites. Also the GWP of site UG was significantly reduced.
However, the site still acted as a C source and showed substantial $N_2O$ emissions, indicating that
rewetting had not yet restored the natural peatland functioning as a sink for C and a negligible source
for $N_2O$. Restoration progress could be promoted by a year-round stabilization of GWL close to the soil
surface. In the current state, a mulching of vegetation in summer might increase total annual
productivity by avoiding early senescence and thus reduce C losses.
Arable forage production (site AR) did not induce higher C losses compared to intensive grassland
management and only showed a significantly higher GWP than the wet grassland site (GW) as the
influence of drainage intensity was of dominating importance. However, interannual on-site variability
was additionally affected by management and climatic factors. The beginning of growing season was
identified as a critical period, with higher $CO_2$ losses occurring with an early start of vegetation period.
Yield related GHG emissions increased with increasing drainage and land use intensity in the order
GW, GM and AR, with a significant difference between sites GW and AR.
As arable cropping was associated with a high uncertainty of yield and a high GWP, this type of
management was identified as unsuitable for forage production on fen soils. The wet grassland site
(GW) realized lowest yield related emissions due to a significantly lower GWP in combination with a
non-significantly reduced energy yield compared to sites GM and AR. Thus, this study demonstrated
that there is huge potential for GHG mitigation in intensively utilized peatland areas of northern
Germany which could be realized without eliminating traditional forage production. Reducing the land
use intensity (low N fertilization, late first cut) of increasingly inundating areas as a consequence of peat



loss, could further enhance GHG mitigation and additionally promote nature conservation purposes
(particularly meadow bird protection).
**Acknowledgements**
This study was funded by the *Gesellschaft für Energie und Klima Schleswig-Holstein* (EKSH) limited
liability company as well as the former *Innovationsstiftung Schleswig-Holstein* (ISH) foundation and
the Ministry of Agriculture, the Environment and Rural Areas of the Federal State of Schleswig-
Holstein (MLUR). Their financial support is grateful acknowledged. The selection of study sites was
supported by the State Office of Agriculture, the Environment and Rural Areas (LLUR). We thank the
three farmers of the study sites for their cooperation and the foundation for nature conservation (*Stiftung*
*Naturschutz*) of Schleswig-Holstein for the permission to conduct our measurements in a nature
conservation area. Further thanks go to Howard Skinner for reviewing the manuscript in terms of
linguistic issues and Mario Hasler for his advice on the statistical analyses.

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

Methods for determining emission factors for the use of peat and peatlands - flux measurements and
modelling, Boreal Environ Res, 12, 85-100, 2007b.
Arneth, A., Kurbatova, J., Kolle, O., Shibistova, O. B., Lloyd, J., Vygodskaya, N. N., and Schulze, E.-
D.: Comparative ecosystem–atmosphere exchange of energy and mass in a European Russian and a
central Siberian bog II. Interseasonal and interannual variability of $CO_2$ fluxes, Tellus, 54B, 514-530,
doi:10.1034/j.1600-0889.2002.01349.x, 2002.
Augustin, J. and Joosten, H.: Peatland rewetting and the greenhouse effect, Int Mire Conserv Group
Newsl, 3, 29-30, 2007.



Aurela, M., Riutta, T., Laurila, T., Tuovinen, J.-P., Vesala, T., Tuittila, E.-S., Rinne, J., Haapanala, S.,
and Laine, J.: $CO_2$ exchange of a sedge fen in southern Finland - the impact of a drought period, Tellus,
59B, 826–837, doi:10.1111/j.1600-0889.2007.00309.x, 2007.
Baker, C. W. and Barnes, R.: The application of near infra-red spectrometry to forage evaluation in the
agricultural developement and advisory service, in: Wiseman, J. and Cole, D. J. A. (eds.): Feedstuff
evaluation, Butterworths, London, 337-351, 1990.
Beetz, S., Liebersbach, H., Glatzel, S., Jurasinski, G., Buczko, U., and Höper, H.: Effects of land use
intensity on the full greenhouse gas balance in an Atlantic peat bog, Biogeosciences, 10, 1067-1082,
doi:10.5194/bg-10-1067-2013, 2013.
Bellisario, L. M., Moore, T. R., and Bubier, J. L.: Net ecosystem $CO_2$ exchange in a boreal peatland,
northern Manitoba, Écoscience, 5, 534-541, 1998.
Beyer, C. and Höper, H.: Greenhouse gas emissions from rewetted bog peat extraction sites and a
Sphagnum cultivation site in Northwest Germany, Biogeosciences Discuss, 11, 4493-4530,
doi:10.5194/bgd-11-4493-2014, 2014.
Blankenburg, J., Hennings, H. H., and Schmidt, W.: Bodenphysikalische Eigenschaften und
Wiedervernässung, in: Kratz, R. and Pfadenhauer, J. (eds.): Ökosystemmanagement für Niedermoore -
Strategien und Verfahren zur Renaturierung, Eugen Ulmer, Stuttgart, 81-91, 2001.
Blume, H.-P. and Brümmer, G.: Agriculture, landscapes and soils of Schleswig-Holstein, Mitt Dtsch
Bodenkdl Ges, 51, 3-14, 1986.
Bouwman, A. F., Boumans, L. M., and Batjes, N. H.: Emissions of $N_2O$ and NO from fertilized fields:
summary of available measurement data, Global Biogeochem Cy, 16, 1-6, doi:10.1029/2001GB001811,

22  2002.

Bretz, F., Hothorn, T., and Westfall, P.: Multiple comparisons using R, Chapman & Hall, CRC Press,
London, 2011.
Bubier, J. L. and Moore, T. R.: An ecological perspective on methane emissions from northern
wetlands, Trends Ecol Evol, 9, 460-464, doi:10.1016/0169-5347(94)90309-3, 1994.
Bubier, J. L., Crill, P. M., Moore, T. R., Savage, K., and Varner, R. K.: Seasonal patterns and controls
on net ecosystem $CO_2$ exchange in a boreal peatland complex, Global Biochem Cy, 12, 703-714,
doi:10.1029/98GB02426, 1998.
Bubier, J. L., Frolking, S., Crill, P. M., and Linder, E.: Net ecosystem productivity and its uncertainty in
a diverse boreal peatland, J Geophys Res, 104, 27683-27692, doi:10.1029/1999JD900219, 1999.





Byrne, K. A., Chojnicki, B., Christensen, T. R., Drösler, M., Freibauer, A., Friborg, T., Frolking, S.,
Lindroth, A., Mailhammer, J., Malmer, N., Selin, P.; Turunen, J., Valentini, R., and Zetterberg, L.: EU
peatlands: Current carbon stocks and trace gas fluxes, Carbo-Europe-GHG Concerted Action-Synthesis
of the European Greenhouse Gas Budget, Report 4, Lund, 2004.
Chapin, F. S., Woodwell, G. M., Randerson, J. T., Rastetter, E. B., Lovett, G. M., Baldocchi, D. D.,
Clark, D. A., Harmon, M. E., Schimel, D. S., Valentini, R., Wirth, C., Aber, J. D., Cole, J. J., Goulden,
M. L., Harden, J. W., Heimann, M., Howarth, R. W.,  Matson, P. A., McGuire, A. D., Melillo, J. M.,
Mooney, H. A., Neff, J. C., Houghton, R. A., Pace, M. L., Ryan, M. G., Running, S. W., Sala, O. E.,
Schlesinger, W. H., and Schulze, E.-D.: Reconciling carbon cycle concepts, terminology, and methods,
Ecosystems, 9, 1041-1050, doi:10.1007/s10021-005-0105-7, 2006.
Chapuis-Lardy, L., Wrage, N., Metay, A., Chotte, J.-L., and Bernoux, M.: Soils, a sink for $N_2O$? A
review, Glob Change Biol, 13, 1-17, doi:10.1111/j.1365-2486.2006.01280.x, 2007.
Chatskikh, D., Olesen, J. E., Hansen, E. M., Elsgaard, L. and Petersen, B. M.: Effects of reduced tillage
on net greenhouse gas fluxes from loamy sand soil under winter crops in Denmark, Agr Ecosys
Environ, 128, 117-126, doi:10.1016/j.agee.2008.05.010, 2008
Couwenberg, J.: Methane emissions from peat soils (organic soils, histosols) - Facts, mrv-ability,
emission factors, Wetlands International, Ede, 2009a.
Couwenberg, J.: Emission factors for managed peat soils (organic soils, histosols) - An analysis of
IPCC default values, Wetlands International, Ede, 2009b.
Couwenberg, J.: Greenhouse gas emissions from managed peat soils: Is the IPCC reporting guidance
realistic?, Mires and Peat, 8, 1-10, 2011.
Couwenberg, J., Thiele, A., Tanneberger, F., Augustin, J., Bärisch, S., Dubovik, D., Liashchynskaya,
N., Michaelis, D., Minke, M., Skuratovich, A., and Joosten, H.: Assessing greenhouse gas emissions
from peatlands using vegatation as a proxy, Hydrobiologia, 674, 67-89, doi:10.1007/s10750-011-0729-
x, 2011.
Davidson, E. A., Savage, K., Verchot, L. V., and Navarro, R.: Minimizing artifacts and biases in
chamber-based measurements of soil respiration, Agr Forest Meteorol, 113, 21-37, doi:10.1016/S0168-

28  1923(02)00100-4, 2002.

Dinsmore, K. J., Skiba, U. M., Billett, M. F., and Rees, R. M.: Effect of water table on greenhouse gas
emissions from peatland mesocosms, Plant Soil, 318, 229-242, doi:10.1007/s11104-008-9832-9, 2009.



Dixon, R. K., Brown, S., Houghton, R. A., Solomon, A. M., Trexler, M. C., and Wisniewski, J.: Carbon
pools and flux of global forest ecosystems, Science, 263, 185-190, doi:10.1126/science.263.5144.185,

3  1994.

Dowrick, D. J., Hughes, S., Freeman, C., Lock, M. A., Reynolds, B., and Hudson, J. A.: Nitrous oxide
emissions from gully mire in mid-Wales, UK, under simulated summer drought, Biogeochemistry, 44,
151-162, doi:10.1023/A:1006031731037, 1999.
Drösler, M.: Trace gas exchange and climatic relevance of bog ecosystems, Southern Germany, Ph.D.
thesis, Technische Universität München, Germany, 2005.
Drösler, M., Freibauer, A., Christensen, T. R., and Friborg, T.: Observations and status of peatland
greenhouse gas emissions in Europe, in: Dolman, A. J., Valentini, R., and Freibauer, A. (eds.): The
continental-scale greenhouse gas balance of Europe, Springer, New York, 243-261, 2008.
Drösler, M., Adelmann, W., Augustin, J., Gergmann, L., Beyer, C., Chojnicki, B., Förster, C.,
Freibauer, A., Giebels, M., Görlitz, S., Höper, H., Kantelhardt, J., Liebersbach, H., Hahn-Schöfl, M.,
Minke, M., Petschow, U., Pfadenhauer, J., Schaller, L., Schägner, P., Sommer, M., Thuille, A., and
Wehrhan, M.: Klimaschutz durch Moorschutz, Schlussbericht des Vorhabens "Klimaschutz -
Moorschutzstrategien", 2006 – 2013, 2013.
Dunfield, P., Knowles, R., Dumont, R., and Moore, T. R.: Methane production and consumption in
temperate and subarctic peat soils: Response to temperature and pH, Soil Biol Biochem, 25, 321-326,
doi:10.1016/0038-0717(93)90130-4, 1993.
Elsgaard, L., Görres, C.-M., Hoffmann, C. C., Blicher-Mathiesen, G., Schelde, K., and Petersen, S. O.:
Net ecosystem exchange of $CO_2$ and carbon balance for eight temperate organic soils under agricultural
management, Agr Ecosys Environ, 162, 52-67, doi:10.1016/j.agee.2012.09.001, 2012.
Estop-Aragonés, C. and Blodau, C.: Effects of experimental drying intensity and duration on respiration
and methane production recovery in fen peat incubations, Soil Biol Biochem, 47, 1-9,
doi:10.1016/j.soilbio.2011.12.008, 2012.
Estop-Aragonés, C., Knorr, K.-H., and Blodau, C.: Controls on in situ oxygen and dissolved inorganic
carbon dynamics in peats of a temperate fen, J Geophys Res, 117, G02002, doi:10.1029/2011JG001888,

28  2012.

FAO: World reference base for soil resources 2006 – a framework for international classification,
correlation and communication, FAO, Rome, 2006.
Farquharson, R. and Baldock, J.: Concepts in modelling $N_2O$ emissions from land use, Plant Soil, 309,
147-167, doi:10.1007/s11104-007-9485-0, 2008.

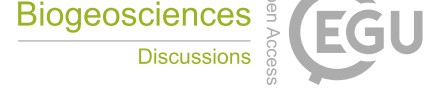
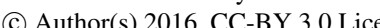
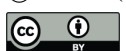

Firestone, M. K., Firestone, R. B., and Tiedje, J. M.: Nitrous oxide from soil denitrification: Factors
controlling its biological production, Science, 208, 749-751, doi:10.1126/science.208.4445.749, 1980.
Flessa, H., Wild, U., Klemisch, M., and Pfadenhauer, J.: Nitrous oxide and methane fluxes from organic
soils under agriculture, Eur J Soil Sci, 49, 327-335, doi:10.1046/j.1365-2389.1998.00156.x, 1998.
GfE: New equations for predicting metabolizable energy of grass and maize products for ruminants, P
Soc Nutr Physiol, 17, 191-197, 2008.
Glatzel, S. and Well, R.: Evaluation of septum-capped vials for storage of gas samples during air
transport, Environ Monit Assess, 136, 307-311, doi:10.1007/s10661-007-9686-2, 2008.
Goldberg, S. D., Knorr, K.-H., Blodau, C., Lischeid, G., and Gebauer, G.: Impact of altering the water
table height of an acidic fen on $N_2O$ and NO fluxes and soil concentrations, Glob Change Biol, 16, 220-
233, doi:10.1111/j.1365-2486.2009.02015.x, 2010.
Gorham, E. and Rochefort, L.: Petaland restoration: A brief assessment with special reference to
Sphagnum bogs, Wetl Ecol Manag, 11, 109-119, doi:10.1023/A:1022065723511, 2003.
Görres, C.-M., Conrad, R., and Petersen, S. O.: Effect of soil properties and hydrology on Archeal
community composition in three temperate grasslands on peat, FEMS Microbiol Ecol, 85, 227-240,
doi:10.1111/1574-6941.12115, 2013.
Grant, R. F., Pattey, E., Goddard, T. W., Kryzanowski, L. M., and Puurveen, H.: Modeling the effects
of fertilizer application rate on nitrous oxide emissions, Soil Sci Soc Am J, 70, 235-248,
doi:10.2136/sssaj2005.0104, 2006.
Griffis, T. J., Rouse, W. R., and Waddington, J. M.: Interannual variability of net ecosystem $CO_2$
exchange at a subarctic fen, Global Biogeochem Cy, 14, 1109-1121, doi:10.1029/1999GB001243,

22 2000.

Grosse-Brauckmann, G.: Moore und Moor-Naturschutzgebiete in Deutschland – eine
Bestandsaufnahme, Telma, 27, 183-215, 1997.
Hahn-Schöfl, M., Zak, D., Minke, M., Gelbrecht, J., Augustin, J., and Freibauer, A.: Organic sediment
formed during inundation of a degraded fen grassland emits large fluxes of $CH_4$ and $CO_2$,
Biogeosciences, 8, 1539-1550, doi:10.5194/bg-8-1539-2011, 2011.
Herrmann, A., Claus, S., Loges, R., Kluß, C., and Taube, F.: Can arable forage production be intensified
sustainably? A case study from northern Germany, Crop Pasture Sci, 65, 538-549,
doi:10.1071/CP13362, 2014.



Hopkins, A. and Green, J. O.: The effect of soil fertility and drainage on sward changes, in: Charles, A. H. and Haggar, R. J. (eds.): Changes in sward composition and productivity, British Grassland Society, Hurley, 115-129, 1979.

Höper, H.: Carbon and nitrogen mineralisation rates of fens in Germany used for agriculture. A review, in: Broll, G., Merbach, W, and Pfeiffer, E.-M. (eds.): Wetlands in Central Europe - soil organisms, soil ecological processes and trace gas emissions, Springer, Berlin, 149-164, 2002.

Höper, H., Augustin, J., Cagampan, J. P., Drösler, M., Lundin, L., Moors, E., Vasander, H, Waddington, J. M., and Wilson, D.: Restoration of peatlands and greenhouse gas balances, in: Strack, M. (ed.): Peatlands and climate change, International Peat Society, Jyväskylä, 182-210, 2008.

Hutchinson, G. L. and Mosier, A. R.: Improved soil cover method for field measurement of nitrous oxide fluxes, Soil Sci Soc Am J, 45, 311-316, doi:10.2136/sssaj1981.03615995004500020017x, 1981.

IPCC: Climate change 2007: The physical science basis. Contribution of working group I to the fourth assessment report of the International Panel on Climate Change, Solomon, S., Qin, D., Manning, M., Chen, Z., Marquis, M., Averyt, K. B., Tignor, M., and Miller, H. L. (eds.), Cambridge University Press, Cambridge, New York, 2007.

IPCC: 2013 Supplement to the 2006 IPCC Guidelines for National Greenhouse Gas Inventories: Wetlands, Hirashi, T., Krug, T., Tanabe, K., Srivastava, N., Baasansuren, J., Fukuda, M., and Troxler, T. G. (eds.), IPCC, Switzerland, 2014.

Janssens, W.: Defining the vegetation period by temperature sums, Proc 7th Conf Biometeorology, Freiburg, 312-318, 2010.

Joosten, H.: The Global Peatland $CO_2$ Picture - Peatland status and emissions in all countries of the world, Wetlands International, Ede, 2009.

Joosten, H. and Clarke, D.: Wise use of mires and peatlands - Background and principles including a framework for decision making, International Mire Conservation Group and International Peat Society, Saarijärvi, 2002.

Jørgensen, C. J. and Elberling, B.: Effects of flooding-induced $N_2O$ production, consumption and emission dynamics on the annual $N_2O$ emission budget in wetland soil, Soil Biol Biochem, 53, 9-17, doi:10.1016/j.soilbio.2012.05.005, 2012.

Kasimir-Klemdtsson, A., Klemdtsson, L., Berglund, K., Martikainen, P., Silvola, J., and Oenema, O.: Greenhouse gas emissions from farmed organic soils: A review, Soil Use Manage, 13, 245-250, doi:10.1111/j.1475-2743.1997.tb00595.x, 1997.





Knorr, K.-H., Glaser, B., and Blodau, C.: Fluxes and 13C isotopic composition of dissolved carbon and
pathways of methanogenesis in a fen soil exposed to experimental drought, Biogeosciences, 5, 1457-
1473, doi:10.5194/bg-5-1457-2008, 2008.
Koops, J. G., van Beusichem, M. L., and Oenema, O.: Nitrogen loss from grassland on peat soils
through nitrous oxide production, Plant Soil, 188, 119-130, doi:10.1023/A:1004252012290, 1997.
Koponen, H. T., Escude Duran, C., Maljanen, M., Hytönen, J., and Martikainen, P. J.: Temperature
responses of NO and $N_2O$ emissions from boreal organic soil, Soil Biol Biochem, 38, 1779-1787,
doi:10.1016/j.soilbio.2005.12.004, 2006.
Laird, N. M. and Ware, J. H.: Random-effects models for longitudinal data, Biometrics, 38, 963-974,
doi:10.2307/2529876, 1982.
Langeveld, C. A., Segers, R., Dirks, B. O., van den Pol-van Dasselaar, A., Velthof, G. L., and Hensen,
A.: Emissions of $CO_2$, $CH_4$ and $N_2O$ from pasture on drained peat soils in the Netherlands, Eur J Agron,
7, 35-42, doi:10.1016/S1161-0301(97)00036-1, 1997.
Leiber-Sauheitl, K., Fuß, R., Voigt, C., and Freibauer, A.: High $CO_2$ fluxes from grassland on histic
Gleysol along soil carbon and drainage gradients, Biogeosciences, 11, 749-761, doi:10.5194/bg-11-749-
16  2014, 2014.

Leppälä, M., Laine, A. M., Seväkivi, M.-L., and Tuittila, E.-S.: Differences in $CO_2$ dynamics between
successional mire plant communities during wet and dry summers, J Veg Sci, 22, 357–366,
doi:10.1111/j.1654-1103.2011.01259.x, 2011.
Leppelt, T., Dechow, R., Gebbert, S., Freibauer, A., Lohila, A., Augustin, J., Drösler, M., Fiedler, S.,
Glatzel, S., Höper, H., Järveoja, J., Lærke, P. E., Maljanen, M., Mander, Ü., Mäkiranta, P., Minkkinen,
K., Ojanen, P., Regina, K., and Strömgren, M.: Nitrous oxide emission hotspots from organic soils in
Europe, Biogeosciences, 11, 6595-6612, doi:10.5194/bg-11-6595-2014, 2014.
Levy, P. E., Gray, A., Leeson, S. R., Gaiawyn, J., Kelly, M. P. C., Cooper, M. D. A., Dinsmore, K. J.,
Jones, S. K., and Sheppard, L. J.: Quantification of uncertainty in trace gas fluxes measured by the static
chamber method, Eur J Soil Sci, 62, 811-821, doi:10.1111/j.1365-2389.2011.01403.x, 2011
Lloyd, J. and Taylor, J. A.: On the temperature dependence of soil respiration, Funct Ecol, 8, 315-323,
28  1994.

Maljanen, M., Liikanen, A., Silvola, J., and Martikainen, P. J.: Methane fluxes on agricultural and
forested boreal organic soils, Soil Use Manage, 19, 73-79, doi:10.1111/j.1475-2743.2003.tb00282.x,
2003a.




Maljanen, M., Liikanen, A., Silvola, J., and Martikainen, P. J.: Nitrous oxide emissions from boreal
organic soil under different land-use, Soil Biol Biochem, 35, 1-12, doi:10.1016/S0038-0717(03)00085-
3, 2003b.
Maljanen, M., Komulainen, V.-M., Hytönen, J., Martikainen, P. J., and Laine, J.: Carbon dioxide,
nitrous oxide and methane dynamics in boreal organic agricultural soils with different soil
characteristics, Soil Biol Biochem, 36, 1801-1808, doi:10.1016/j.soilbio.2004.05.003, 2004.
Maljanen, M., Sigurdsson, B. D., Gudmundsson, J., Oskarsson, H., Huttunen, J. T., and Martikainen, P.
J.: Greenhouse gas balances of managed peatlands in the Nordic countries - present knowledge and
gaps, Biogeosciences, 7, 2711-2738, doi:10.5194/bg-7-2711-2010, 2010.
Martikainen, P. J., Nykänen, H., Crill, P., and Silvola, J.: Effect of a lowered water table on nitrous
oxide fluxes from northern peatlands, Nature, 366, 51-53, doi:10.1038/366051a0, 1993.
McInerney, J. P.: Economic aspects of grassland production and utilization, in: Hopkins, A. (ed.):
Grass: Its production and utilization, Blackwell Science Ltd., Oxford, 394-428, 2000.
Michaelis, L. and Menten, M. L.: Die Kinetik der Invertinwirkung, Biochem Z, 49, 333-369, 1913.
Moore, T. R. and Knowles, R.: The influence of water table levels on methane and carbon dioxide
emissions from peatland soils, Can J Soil Sci, 69, 33-38, doi:10.4141/cjss89-004, 1989.
Muhr, J., Höhle, J., Otieno, D. O., and Borken, W.: Manipulative lowering of the water table during
summer does not affect $CO_2$ emissions and uptake in a fen in Germany, Ecol Appl, 21, 391-401,
doi:10.1890/09-1251.1, 2011.
Nykänen, H., Alm, J., Lang, K., Silvola, J., and Martikainen, P. J.: Emissions of $CH_4$, $N_2O$ and $CO_2$
from a virgin fen and a fen drained for grassland in Finland, J Biogeogr, 22, 351-357, 1995.
Nykänen, H., Alm, J., Silvola, J., Tolonen, K., and Martikainen, P. J.: Methane fluxes on boreal
peatlands of different fertility and the effect of long-term experimental lowering of the water table on
flux rates, Global Biogeochem Cy, 12, 53-69, doi:10.1029/97GB02732, 1998.
Oenema, O., de Klein, C., and Alfaro, M.: Intensification of grassland and forage use: Driving forces
and constraints, Crop Pasture Sci, 65, 524-537, doi:10.1071/CP14001, 2014.
Oleszczuk, R., Regina, K., Szajdak, L., Höper, H., and Maryganova, V.: Impacts of agricultural
utilization of peat soils on the greenhouse gas balance, in: Strack, M. (ed.): Peatlands and climate
change, International Peat Society, Jyväskylä, 70-97, 2008.



Parmentier, F. J., van der Molen, M. K., de Jeu, R. A., Hendriks, D. M., and Dolman, A. J.: $CO_2$ fluxes
and evaporation on a peatland in the Netherlands appear not affected by water table fluctuations, Agr
Forest Meteorol, 149, 1201-1208, doi:10.1016/j.agrformet.2008.11.007, 2009.
Parsons, A. J. and Chapman, D. F.: The principles of pasture growth and utilization, in: Hopkins, A.
(ed.): Grass: Its production and utilization, Blackwell Science Ltd., Oxford, 31-89, 2000.
Petersen, S. O., Hoffmann, C. C., Schäfer, C.-M., Blicher-Mathiesen, G., Elsgaard, L., Kristensen, K.,
Larsen, S. E., Torp, S. B., and Greve, M. H.: Annual emissions of $CH_4$ and $N_2O$, and ecosystem
respiration, from eight organic soils in Western Denmark managed by agriculture, Biogeosciences, 9,
403-422, doi:10.5194/bg-9-403-2012, 2012.
Priemé, A. and Christensen, S.: Natural perturbations, drying – wetting and freezing – thawing cycles,
and the emission of nitrous oxide, carbon dioxide and methane from farmed organic soils, Soil Biol
Biochem, 33, 2083-2091, doi:10.1016/S0038-0717(01)00140-7, 2001.
R Core Team: R: A language and environment for statistical computing, R Foundation for Statistical
Computing, URL: http://www.R-project.org/, Wien, 2014.
Regina, K., Syväsalo, E., Hannukkala, A., and Esala, M.: Fluxes of $N_2O$ from farmed peat soils in
Finland, Eur J Soil Sci, 55, 591-599, doi:10.1111/j.1365-2389.2004.00622.x, 2004.
Regina, K., Sheehy, J., and Myllys, M.: Mitigating greenhouse gas fluxes from cultivated organic soils
with raised water table, Mitig Adapt Strateg Glob Change, doi:10.1007/s11027-014-9559-2, 2014.
Renger, M., Wessolek, G., Schwärzel, K., Sauerbrey, R., and Siewert, C.: Aspects of peat conservation
and   water   management.   J   Plant   Nutr   Soil   Sc,   165,   487-493,   doi:10.1002/1522-
2624(200208)165:4<487::AID-JPLN487>3.0.CO;2-C, 2002.
Robertson, G. P., Paul, E. A., and Harwood, R. R.: Greenhouse gases in intensive agriculture:
Contributions of individual gases to the radiative forcing of the atmosphere, Science, 289, 1922-1925,
doi:10.1126/science.289.5486.1922, 2000.
Rohman, K., Jeromin, K., Kieckbusch, J., Koop, B., and Struwe-Juhl, B.: Europäischer Vogelschutz in
Schleswig-Holstein - Arten und Schutzgebiete, Landesamt für Natur und Umwelt Schleswig-Holstein,
Flintbek, 2008.
Roobroeck, D., Brüggemann, N., Butterbach-Bahl, K., and Boeckx, P.: Dynamics of nitrate limitation
on gaseous nitrogen exchanges from pristine peatlands, Geophys Res Abstr, 11, EGU2009-12298-3,
30  2009.



Roulet, N. T., Ash, R., and Quinton, W.: Methane flux from drained northern peatlands: Effect of a
persistant water table lowering on flux, Global Biogeochem Cy, 7, 749-769, doi:10.1029/93GB01931,

3   1993.

Röder, N., Osterburg, B., and Nitsch, N.: Regional differences in the intensity of the utilisation of
organic soils in Germany, in: Pötsch, E. M., Krautzer, B., and Hopkins, A. (eds.): Grassland farming
and land management systems in mountainous regions, Grassl Sci Eur, 16, 347-349, 2011.
Röder, N. and Osterburg, B.: The impact of map and data resolution on the determination of the
agricultural utilisation of organic soils in Germany, Environ Manage, 49, 1150-1162,
doi:10.1007/s00267-012-9849-y, 2012.
Schäfer, C.-M., Elsgaard, L., Hoffmann, C. C., and Petersen, S. O.: Seasonal methane dynamics in three
temperate grasslands on peat, Plant Soil, 357, 339-353, doi:10.1007/s11104-012-1168-9, 2012.
Schmitt, M., Bahn, M., Wohlfahrt, G., Tappeiner, U., and Cernusca, A.: Land use affects the net
ecosystem $CO_2$ exchange and its components in mountain grasslands, Biogeosciences, 7, 2297-2309,
doi:10.5194/bg-7-2297-2010, 2010.
Schrautzer, J., Sival, F., Breuer, M., Runhaar, H., and Fichtner, A.: Characterizing and evaluating
successional pathways of fen degradation and restoration, Ecol Indic, 25, 108-120,
doi:10.1016/j.ecolind.2012.08.018, 2013.
Senbayram, M., Chen, R., Budai, A., Bakken, L., and Dittert, K.: $N_2O$ emission and the $N_2O/(N_2O +$
$N_2)$ product ratio of denitrification as controlled by available carbon substrates and nitrate
concentrations, Agr Ecosys Environ, 147, 4-12, doi:10.1016/j.agee.2011.06.022, 2012.
Silvan, N., Tuittila, E.-S., Kitunen, V., Vasander, H., and Laine, J.: Nitrate uptake by Eriophorum
vaginatum controls $N_2O$ production in a restored peatland, Soil Biol Biochem, 37, 1519-1526,
doi:10.1016/j.soilbio.2005.01.006, 2005.
Smith, P., Martino, D., Cai, Z., Gwary, D., Janzen, H., Kumar, P., McCarl, B., Ogle, S., O'Mara, F.,
Rice, C., Scholes, B., Sirotenko, O., Howden, M., McAllister, T., Pan, G., Romanenkov, V., Schneider,
U., Towprayoon, S., Wattenbach, M., and Smith, J.: Greenhouse gas mitigation in agriculture, Philos T
R Soc B, 363, 789-813, doi:10.1098/rstb.2007.2184, 2008.
Soussana, J.-F., Allard, V., Pilegaard, K., Ambus, P., Amman, C., Campbell, C., Ceschia, E., Clifton-
Brown, J., Czobel, S., Domingues, R., Flechard, C., Fuhrer, J., Hensen, A., Horvath, L., Jones, M.,
Kasper, G., Martin, C., Nagy, Z., Neftel, A., Raschi, A., Baronti, S., Rees, R. M., Skiba, U., Stefani, P.,
Manca, G., Sutton, M., Tuba, Z., and Valentini, R.: Full accounting of the greenhouse gas ($CO_2$, $N_2O$,



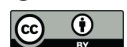

CH$_4$) budget of nine European grassland sites, Agr Ecosys Environ, 121, 121-134, doi:10.1016/j.agee.2006.12.022, 2007.

Timmermann, T., Margóczi, K., Takács, G., and Vegelin, K.: Restoration of peat-forming vegetation by rewetting species poor fen grassland, Appl Veg Sci, 9, 241-250, doi:10.1111/j.1654-109X.2006.tb00673.x, 2006.

Tyson, K. C., Garwood, E. A., Armstrong, A. C., and Scholefield, D.: Effect of field drainage on the growth of herbage and the liveweight gain of grazing beef cattle, Grass Forage Sci, 47, 290-301, doi:10.1111/j.1365-2494.1992.tb02273.x, 1992.

UBA: National Inventory Report for the German Greenhouse Gas Inventory 1990 – 2012, Submission under the United Nations Framework Convention on Climate Change and the Kyoto-Protocol 2014, UBA, Dessau-Roßlau, 2014.

van Beek, C. L., Pleijter, M., Jacobs, C. M., Velthof, G. L., van Groenigen, J. W., and Kuikman, P. J.: Emissions of N$_2$O from fertilized and grazed grassland on organic soil in relation to groundwater level, Nutr Cycl Agroecosys, 86, 331-340, doi:10.1007/s10705-009-9295-2, 2010.

van Beek, C. L., Pleijter, M., and Kuikman, P. J.: Nitrous oxide emissions from fertilized and unfertilized grasslands on peat soil, Nutr Cycl Agroecosys, 89, 453-461, doi:10.1007/s10705-010-9408-y, 2011.

van den Pol-van Dasselaar, A., van Beusichem, M. L., and Oenema, O.: Effects of grassland management on the emission of methane from intensively managed grasslands on peat soil, Plant Soil, 189, 1-9, doi:10.1023/A:1004219522404, 1997.

van den Pol-van Dasselaar, A., van Beusichem, M. L., and Oenema, O.: Determinants of spatial variability of methane emissions from wet grasslands on peat soil, Biogeochemistry, 44, 221-237, doi:10.1023/A:1006009830660, 1999.

van der Weerden, T. J., Clough, T. J., and Styles, T. M.: Using near continuous measurements of N$_2$O emissions from urine-affected soil to guide manual gas sampling regimes, New Zeal J Agr Res, 56, 60-76, doi:10.1080/00288233.2012.747548, 2013.

van Groenigen, J. W., Velthof, G. L., Oenema, O., van Groenigen, K. J., and van Kessel, C.: Towards an agronomic assessment of N$_2$O emissions: A case study for arable crops, Eur J Soil Sci, 61, 903-913, doi:10.1111/j.1365-2389.2009.01217.x, 2010.

Velthof, G. L. and Oenema, O.: Nitrous oxide fluxes from grassland in the Netherlands: I. Statistical analysis of flux-chamber measurements, Eur J Soil Sci, 46, 533-540, 10.1111/j.1365-2389.1995.tb01349.x, 1995a.

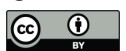



Velthof, G. L. and Oenema, O.: Nitrous oxide fluxes from grassland in the Nethetlands: II. Effects of soil type, nitrogen fertilizer application and grazing, Eur J Soil Sci, 46, 541-549, 10.1111/j.1365-2389.1995.tb01350.x, 1995b.

Verbeke, G. and Molenberghs, G.: Linear mixed models for longitudinal data, Springer, New York, 2000.

Waddington, J. M. and Roulet, N. T.: Atmosphere-wetland carbon exchanges: Scale dependency of $CO_2$ and $CH_4$ exchange on the developmental topography of a peatland, Global Biogeochem Cy, 10, 233-245, doi:10.1029/95GB03871, 1996.

Weißbach, F., Schmidt, L., and Kuhla, S.: Simplified method for calculation of NEL from metabolizable energy, Proc Soc Nutr Physiol, 5, 117, 1996.

Wohlfahrt, G., Anderson-Dunn, M., Bahn, M., Balzarolo, M., Berninger, F., Campbell, C., Carrara, A., Cescatti, A., Christensen, T., Dore, S., Eugster, W., Friborg, T., Furger, M., Gianelle, D., Gimeno, C., Hargreaves, H., Hari, P., Haslwanter, A., Johansson, T., Marcolla, B., Milford, C., Nagy, Z., Nemitz, E., Rogiers, N., Sanz, M. J., Siegwolf, R. T. W., Susiluoto, S., Sutton, M., Tuba, Z., Ugolini, F., Valentini, R., Zorer, R., and Cernusca, A.: Biotic, abiotic, and management controls on the net ecosystem $CO_2$ exchange of European mountain grassland ecosystems, Ecosystems, 11, 1338–1351, doi:10.1007/s10021-008-9196-2, 2008a.

Wohlfahrt, G., Hammerle, A., Haslwanter, A., Bahn, M., Tappeiner, U., and Cernusca, A.: Seasonal and inter-annual variability of the net ecosystem $CO_2$ exchange of a temperate mountain grassland: Effects of weather and management, J Geophys Res, 113, doi:10.1029/2007JD009286, 2008b.

Yrjälä, K., Tuomivirta, T., Juottonen, H., Putkinen, A., Lappi, K., Tuittila, E.-S., Penttilä, T., Minkkinen, K., Laine, J., Peltoniemi, K., and Fritze, H.: $CH_4$ production and oxidation processes in a boreal fen ecosystem after long-term water table drawdown, Global Change Biol, 17, 1311–1320, doi:10.1111/j.1365-2486.2010.02290.x, 2011.

Zak, D., Augustin, J., Trepel, M., and Gelbrecht, J.: Strategies for fen restoration and avoiding conflicts in the context of water, climate and nature protection in the lowlands of NE Germany, Telma, Supplement 4, 133-150, 2011.





Table 1. Soil and land use characterization of the experimental sites (UG: unutilized grassland, GW: grassland 'wet', GM:
grassland 'moist', AR: arable land). Numbers in brackets represent standard deviation.

| Site | UG (1 ha) | GW (3 ha) | GM (3.5 ha) | AR (2.2 ha) |
|---|---|---|---|---|
| Peat depth (cm) | 180 | 420 | 360 | 280 |
| $C_{org}$ (%)[a] | 35.0 (2.6) | 37.4 (3.9) | 17.9 (2.9) | 13.3 (1.9) |
| C/N[a] | 17.7 (1.0) | 15.7 (0.6) | 12.4 (0.4) | 12.2 (0.2) |
| Ash (%)[a] | 36.8 (11.7) | 33.6 (6.3) | 68.7 (2.3) | 74.0 (4.2) |
| Bulk density (g cm$^{-3}$)[a] | 0.20 (0.05) | 0.32 (0.07) | 0.54 (0.08) | 0.67 (0.09) |
| C stock (Mg ha$^{-1}$)[a] | 215 (57) | 361 (82) | 289 (45) | 266 (38) |
| $N_{min}$ (kg ha$^{-1}$)[b] | 20.8 (8.8) | 44.7 (22.7) | 73.1 (37.3) | 65.3 (31.4) |
| $NO_3$-N/$NH_4$-N[b] | 0.10 (0.16) | 0.25 (0.27) | 0.67 (0.61) | 2.55 (3.34) |
| Soil moisture (kg kg$^{-1}$)[b] | 2.84 (0.44) | 2.36 (0.61) | 1.15 (0.30) | 0.79 (0.16) |
| pH ($CaCl_2$)[c] | 4.58 (0.13) | 4.41 (0.18) | 5.06 (0.13) | 5.31 (0.29) |
| Groundwater level (cm)[d] | -10.9 (3.5) | -21.4 (4.6) | -33.0 (9.4) | -39.4 (4.2) |
| Fertilization (kg N ha$^{-1}$ a$^{-1}$)[e] | — | 300 (240 – 400) | 260 (230 – 320) | 150 (130 – 170) |
| Type of fertilizer[f] | — | cattle slurry, CAN, ASN | cattle slurry, CAN | cattle slurry, DAP, CAN |
| Dominant plant species | Purple small-reed (*Calamagrostis canescens*), Reed canary grass (*Phalaris arundinacea*), Common rush (*Juncus effusus*) | Perennial ryegrass (*Lolium perenne*), Rough bluegrass (*Poa trivialis*), Creeping bentgrass (*Agrostis stolonifera*) | Italian ryegrass (*Lolium multiflorum*), Perennial ryegrass (*Lolium perenne*), Rough bluegrass (*Poa trivialis*) | Maize (*Zea mays*), Barley (*Hordeum vulgare*), Wheat (*Triticum aestivum*), Perennial ryegrass (*Lolium perenne*) |

[a] Given values are for 0 – 30 cm soil depth. $C_{org}$ and C/N: mean value from biannual samplings during the period May 2011
– March 2014 (n = 7). Bulk density and C stock: mean value of soil samples taken in May 2013 (n = 4). Ash content: mean
value from samples taken in October 2013 (n = 4). [b] Mean value of mineral nitrogen ($NO_3^-$ and $NH_4^+$) and gravimetric soil
moisture content in 0 – 20 cm soil depth from biweekly samplings during the period April 2011 – April 2014 (n = 73). [c]
Mean value of two samplings in the beginning (May 2011) and in the end (July 2014) of the study (n = 8). [d] Mean value of
linear interpolated weekly measurements in the period April 2011 – March 2014 (n = 4). [e] Sum of applied nitrogen from
organic and mineral fertilizers on average of 2011, 2012 and 2013 and the range between the years. [f] CAN = calcium
ammonium nitrate, ASN = ammonium sulphate nitrate, DAP = diammonium phosphate.



Table 2. Annual budgets of $CO_2$ exchange ($R_{ECO}$, GPP and NEE), $CH_4$ and $N_2O$ fluxes, net ecosystem carbon balance
(NECB) and global warming potential (GWP) for 100 years (IPCC, 2007) for different study periods (each period is April –
March). NECB is calculated from NEE and $CH_4$-C as well as slurry-C and harvested C. The GWP includes $CO_2$-C-
equivalents of NEE, $CH_4$-C, $N_2O$-N, slurry-C and harvested C. Small deviations in NEE are caused by rounding. Values are
annual sums and standard errors (in brackets).

| Site | Period | $R_{ECO}$ | GPP | NEE | $CH_4$ | | $N_2O$ | | NECB | GWP |
|---|---|---|---|---|---|---|---|---|---|---|
| | | (Mg $CO_2$-C ha⁻¹) | | | (kg C ha⁻¹) | (Mg $CO_2$-C-eq ha⁻¹) | (kg N ha⁻¹) | (Mg $CO_2$-C-eq ha⁻¹) | (Mg C ha⁻¹) | (Mg $CO_2$-C-eq ha⁻¹) |
| UG | 2011/12 | — | — | — | 38.6 (10.6) | 0.35 (0.10) | 3.3 (1.2) | 0.43 (0.15) | — | — |
| | 2012/13 | 20.7 (2.3) | -16.8 (3.1) | 3.9 (3.8) | 99.5 (47.8) | 0.91 (0.44) | 2.3 (0.4) | 0.29 (0.05) | 4.0 (3.9) | 5.1 (4.3) |
| | 2013/14 | 17.9 (0.8) | -16.3 (0.9) | 1.7 (1.2) | 27.0 (9.7) | 0.25 (0.09) | 4.5 (1.1) | 0.57 (0.13) | 1.7 (1.2) | 2.5 (1.3) |
| GW | 2011/12 | — | — | — | 6.6 (1.7) | 0.06 (0.02) | 4.7 (1.1) | 0.59 (0.14) | — | — |
| | 2012/13 | 26.6 (0.2) | -17.5 (0.3) | 9.1 (0.4) | 16.5 (2.8) | 0.15 (0.03) | 8.9 (2.0) | 1.14 (0.26) | 13.0 (1.0) | 14.3 (1.2) |
| | 2013/14 | 24.7 (0.8) | -17.8 (0.5) | 6.9 (0.9) | 17.3 (11.7) | 0.16 (0.11) | 5.9 (1.0) | 0.76 (0.13) | 8.3 (1.1) | 9.2 (1.3) |
| GM | 2011/12 | — | — | — | 1.1 (0.6) | 0.01 (0.01) | 11.4 (3.0) | 1.45 (0.38) | — | — |
| | 2012/13 | 29.6 (1.1) | -18.3 (0.4) | 11.4 (1.1) | 1.5 (0.8) | 0.01 (0.01) | 12.5 (1.9) | 1.59 (0.24) | 15.7 (1.3) | 17.3 (1.5) |
| | 2013/14 | 24.9 (1.2) | -13.0 (0.4) | 12.0 (1.3) | 0.2 (0.9) | < 0.01 (0.01) | 19.3 (4.7) | 2.46 (0.60) | 15.7 (1.5) | 18.2 (2.9) |
| AR | 2011/12 | — | — | — | 0.8 (1.8) | 0.01 (0.02) | 20.0 (1.8) | 2.55 (0.26) | — | — |
| | 2012/13 | 24.7 (1.3) | -15.7 (0.9) | 9.0 (1.5) | 4.5 (0.7) | 0.04 (0.01) | 15.3 (1.4) | 1.95 (0.17) | 12.2 (1.8) | 14.2 (1.9) |
| | 2013/14 | 33.3 (2.2) | -22.1 (0.7) | 11.2 (2.3) | 0.2 (0.3) | < 0.01 (< 0.01) | 21.6 (2.1) | 2.76 (0.27) | 17.7 (3.0) | 20.5 (3.3) |

Table 3. **:** Annual yields of dry matter (DM), carbon (C), nitrogen (N) and net energy lactation (NEL) for the three
agricultural utilized study sites and two years. Different capital letters indicate significant differences between the sites for a
particular year. Different lowercase letters indicate significant differences between the years for a particular site (p < 0.05).
Values in brackets are standard errors (n = 3). Crops at site AR were summer barley (2012) and summer wheat with
undersown grass (2013).

| Site | Year | DM ($Mg\ ha^{-1}\ a^{-1}$) | | C ($Mg\ ha^{-1}\ a^{-1}$) | | N ($kg\ ha^{-1}\ a^{-1}$) | | NEL ($GJ\ ha^{-1}\ a^{-1}$) | |
|---|---|---|---|---|---|---|---|---|---|
| | 2012 | 10.7 (1.2) | ABa | 4.9 (0.6) | ABa | 234 (31) | Ba | 63.9 (7.0) | ABa |
| GW | 2013 | 8.2 (0.4) | Aa | 3.7 (0.2) | Aa | 218 (8) | Aa | 53.6 (3.2) | Aa |
| | Mean | 9.5 (0.8) | A | 4.3 (0.4) | A | 226 (15) | AB | 58.7 (3.9) | A |
| | 2012 | 13.1 (0.3) | Bb | 5.9 (0.2) | Bb | 335 (7) | Cb | 78.8 (1.7) | Bb |
| GM | 2013 | 10.0 (0.4) | Ba | 4.5 (0.2) | Ba | 274 (17) | Ba | 66.1 (3.0) | ABa |
| | Mean | 11.5 (0.6) | A | 5.2 (0.3) | A | 305 (13) | B | 72.5 (2.7) | A |
| | 2012 | 8.2 (0.5) | Aa | 3.7 (0.2) | Aa | 107 (11) | Aa | 47.5 (2.6) | Aa |
| AR | 2013 | 14.6 (1.6) | Bb | 6.5 (0.7) | Bb | 296 (34) | ABb | 88.1 (8.2) | Bb |
| | Mean | 11.4 (1.3) | A | 5.1 (0.6) | A | 202 (38) | A | 67.8 (8.2) | A |





1  Table 4. Annual GHG balances ($CO_2$, $CH_4$ and $N_2O$ fluxes, slurry-C and harvested C) of the three agricultural study sites in

2  relation to energy yield (net energy lactation, NEL). Different capital letters indicate significant differences between the sites

3  ($p < 0.05$). Values in brackets are standard errors (n = 3).

| Period | Site | | | | | |
|---|---|---|---|---|---|---|
| | GW | | GM | | AR | |
| | kg $CO_2$-C-eq (GJ NEL)$^{-1}$ | | | | | |
| 2012/13 | 231 (25) | | 220 (5) | | 301 (18) | |
| 2013/14 | 172 (10) | | 276 (12) | | 236 (21) | |
| Mean | 201 (17) | A | 248 (9) | AB | 269 (19) | B |





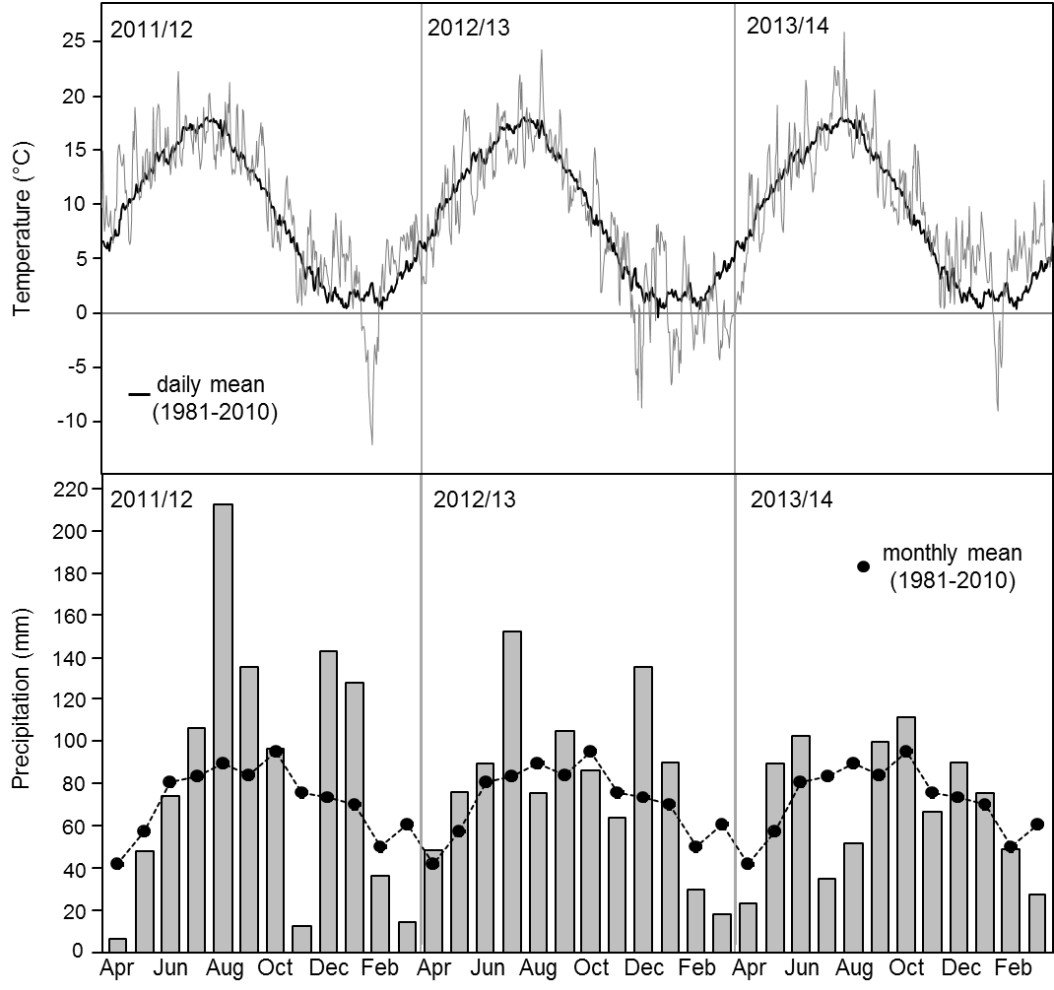

3    Figure 1. Daily mean air temperatures (grey line) and monthly mean precipitation sums (grey bars)

4    during the study period (April 2011 – March 2014) compared to the long-term averages.





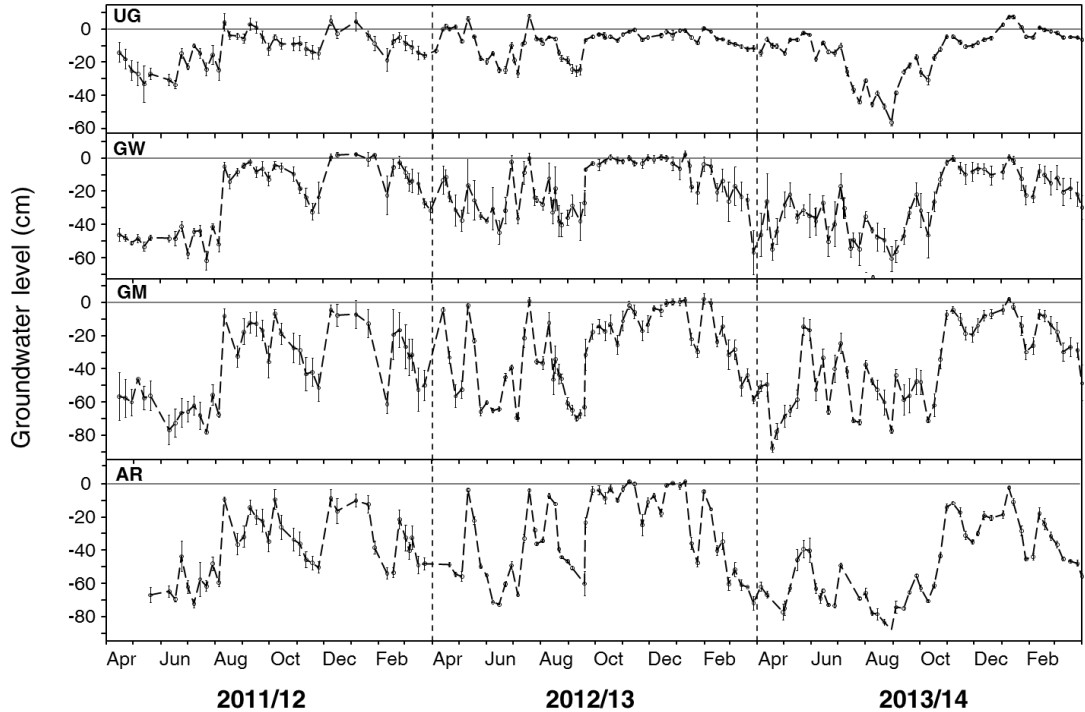

3    Figure 2. Development of groundwater levels (GWLs) at the four study sites during the study period

4    (April 2011 – March 2014). Displayed are mean values ± standard errors of the manually recorded

5    GWLs during gas flux measurements (n = 4).





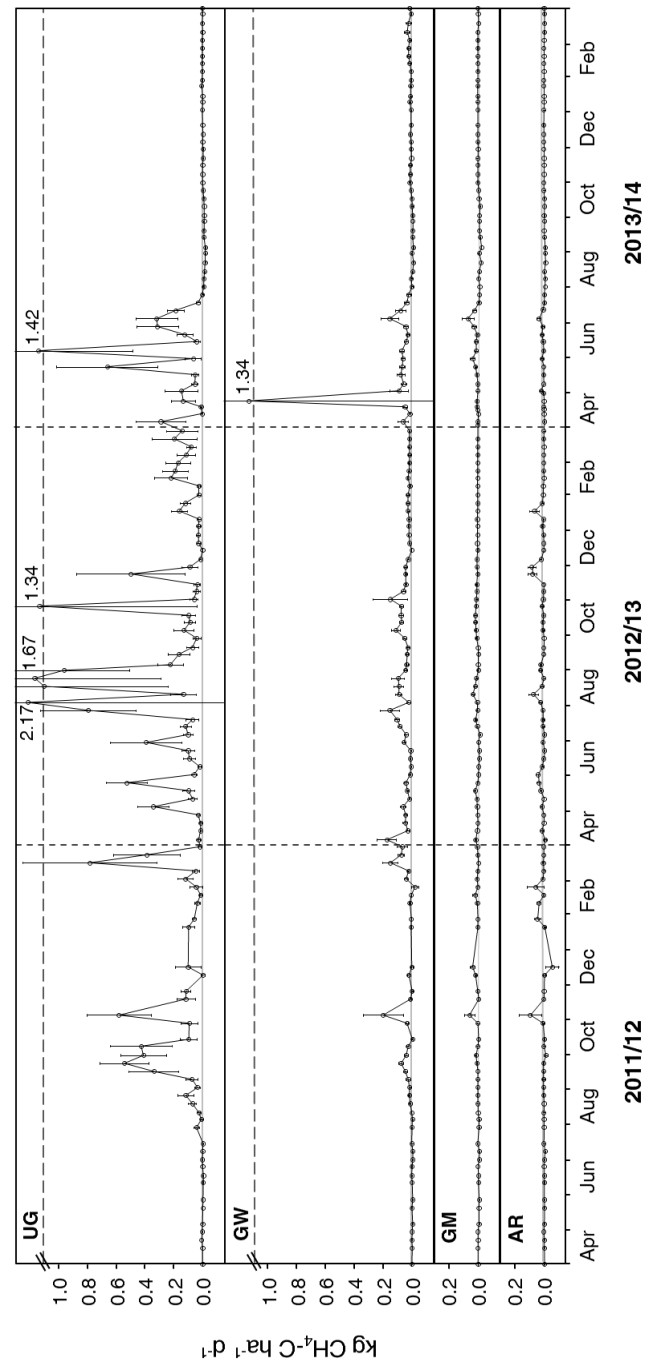

3    Figure 3. CH₄ exchange at the four study sites during the study period (April 2011 − March 2014).

4    Values are displayed as mean ± standard error (n = 8). Note the broken y-axis for sites UG and GW.



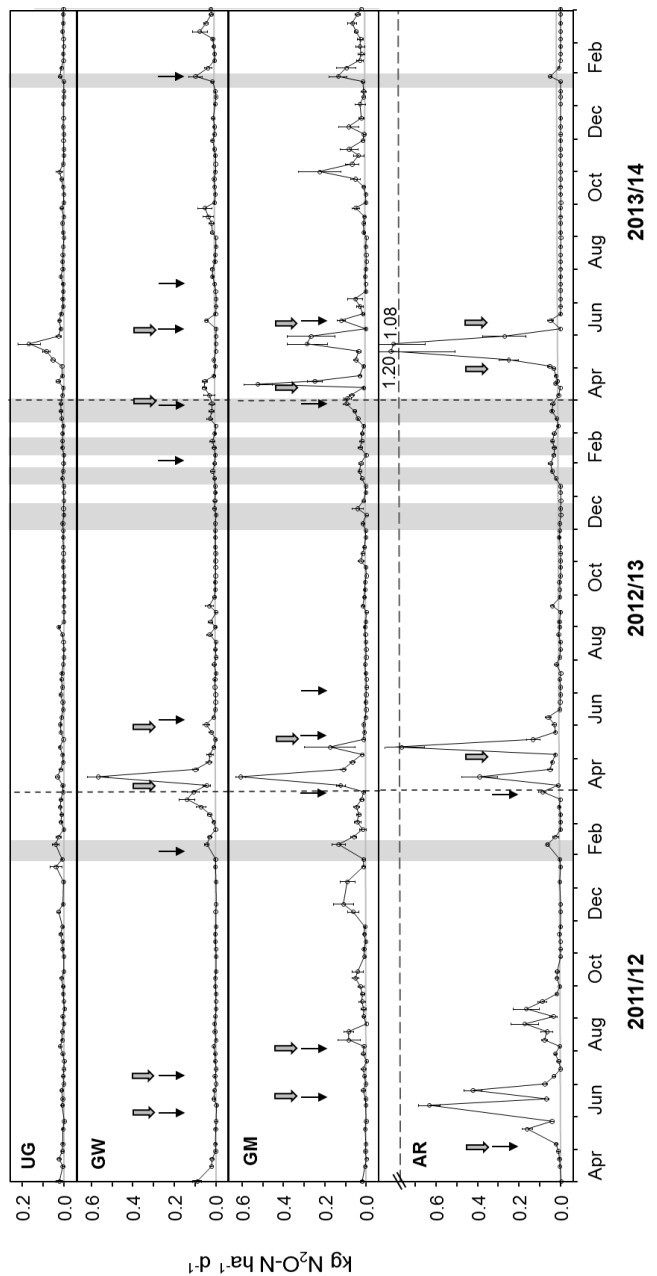

3 Figure 4. N$_2$O exchange at the four study sites during the study period (April 2011 – March 2014).

4 Values are displayed as mean ± standard error (n = 8). Note the broken y-axis for site AR. Arrows

5 indicate applications of slurry (black) and mineral nitrogen fertilizer (grey). Grey background represents

6 periods with mean daily temperatures below 0 °C.



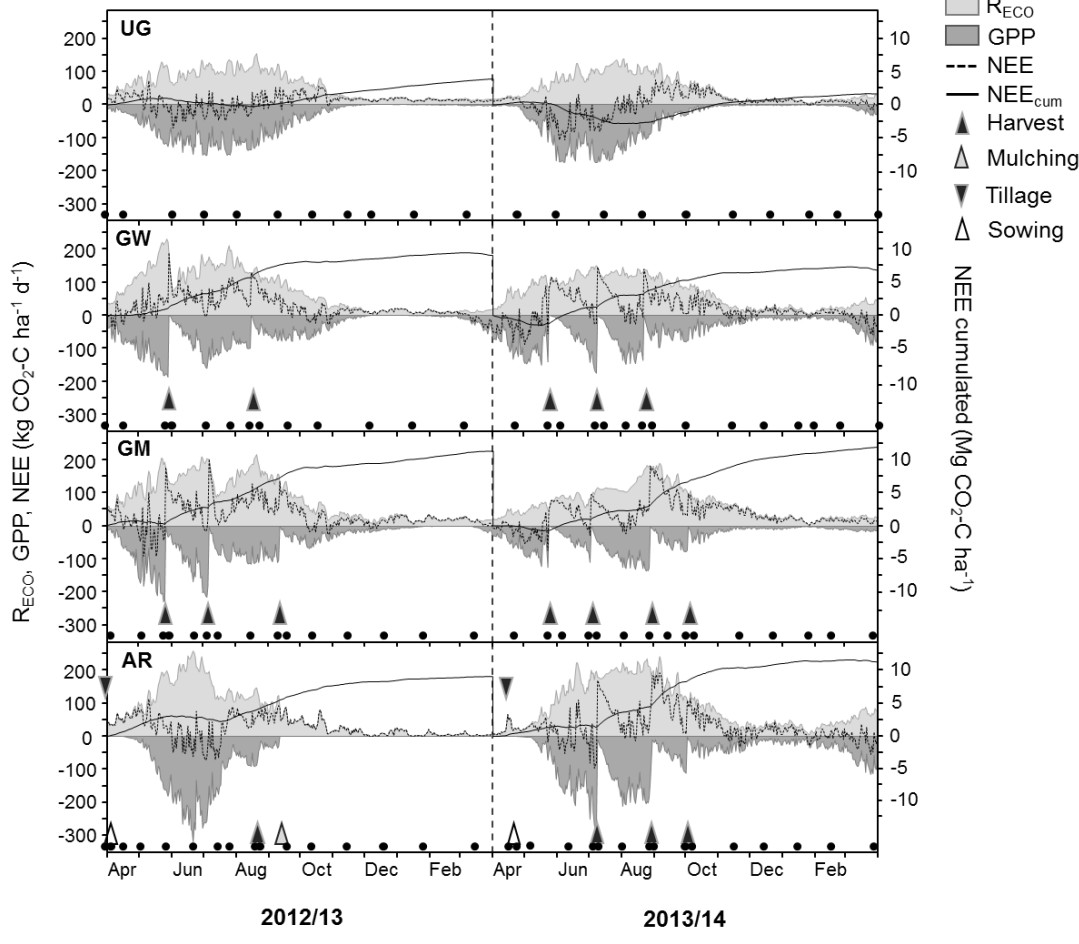

3 Figure 5. CO₂ exchange at the four study sites during a two-year study period (April 2012 – March

4 2014). Values are displayed as daily means of the model output (n = 3). The black continuous line

5 shows the cumulated NEE for one year. The black dots represent CO₂ measurement campaigns.



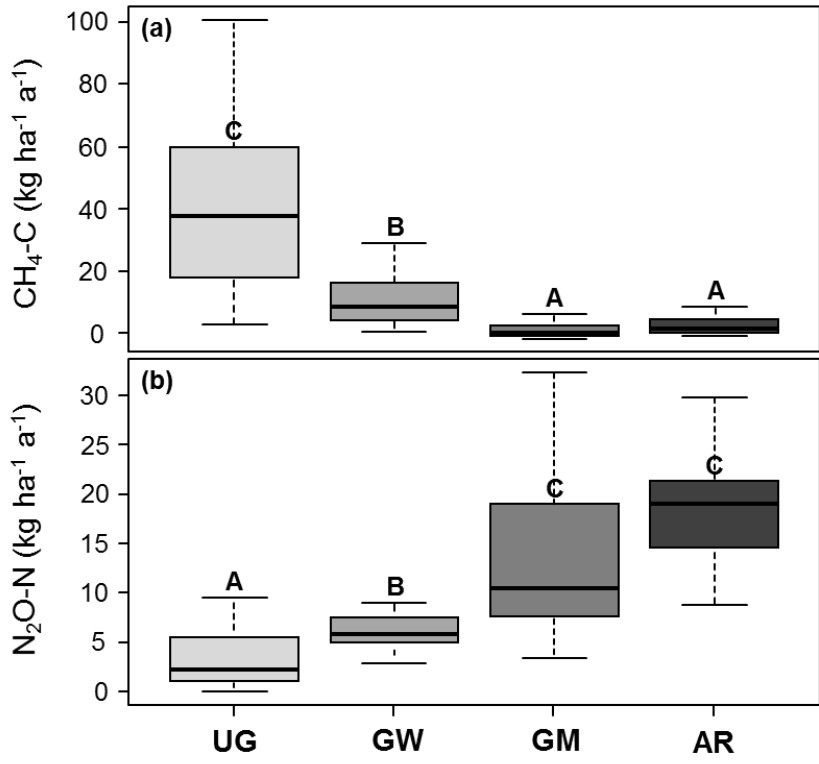

3    Figure 6. Annual emissions of $CH_4$ (a) and $N_2O$ (b) at the four study sites combined for three years

4    (April 2011 – March 2014) of measurement (n = 24). Different capital letters indicate significant

5    differences between the sites (p < 0.05).





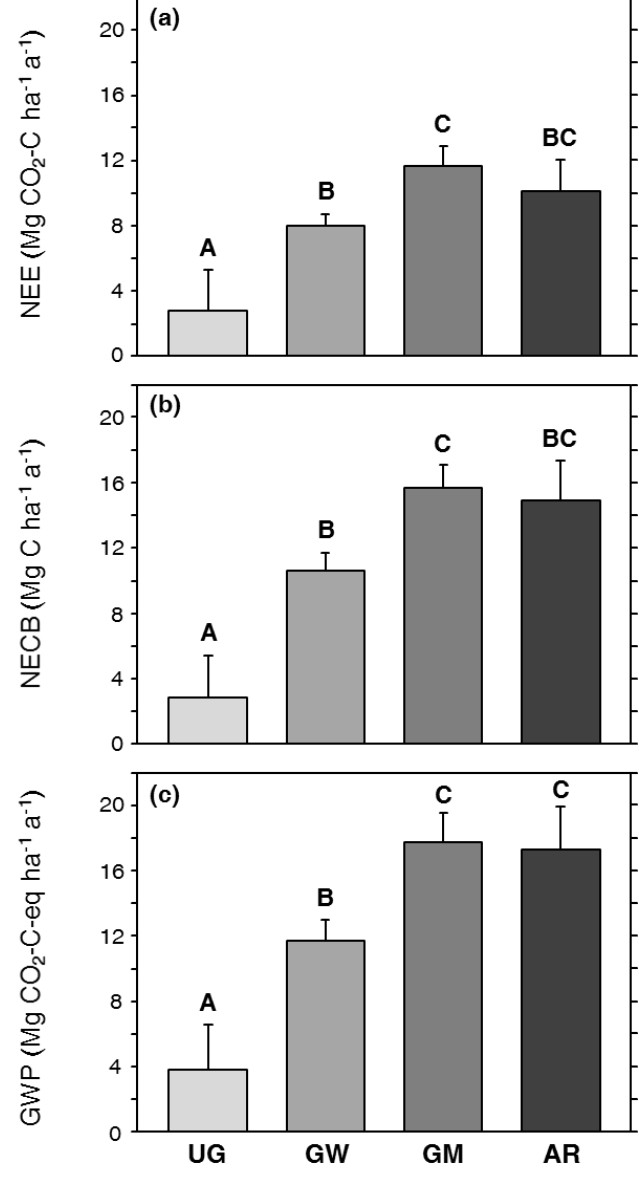

Figure 7. Mean annual budgets of net ecosystem exchange (NEE) of $CO_2$ (a), net ecosystem carbon
balance (NECB) (b), and global warming potential (GWP) (c) at the four study sites for the period April
2012 – March 2014 (n = 6). Error bars represent standard error. Different capital letters indicate
significant differences between the sites (p < 0.05).




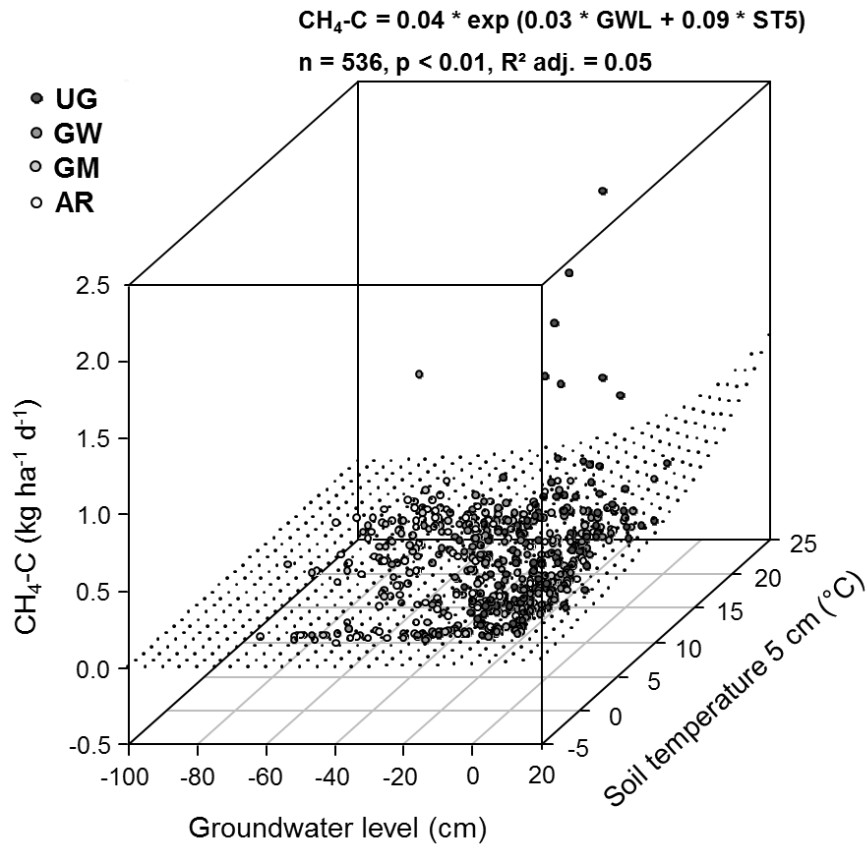

3    Figure 8. Relationship of daily $CH_4$ fluxes to groundwater level and mean daily soil temperature at 5 cm

4    depth. GWL in the equation is groundwater level (cm) and ST5 is soil temperature at 5 cm depth (°C).

5    $R^2$ adjusted was estimated for predicted versus obtained values.





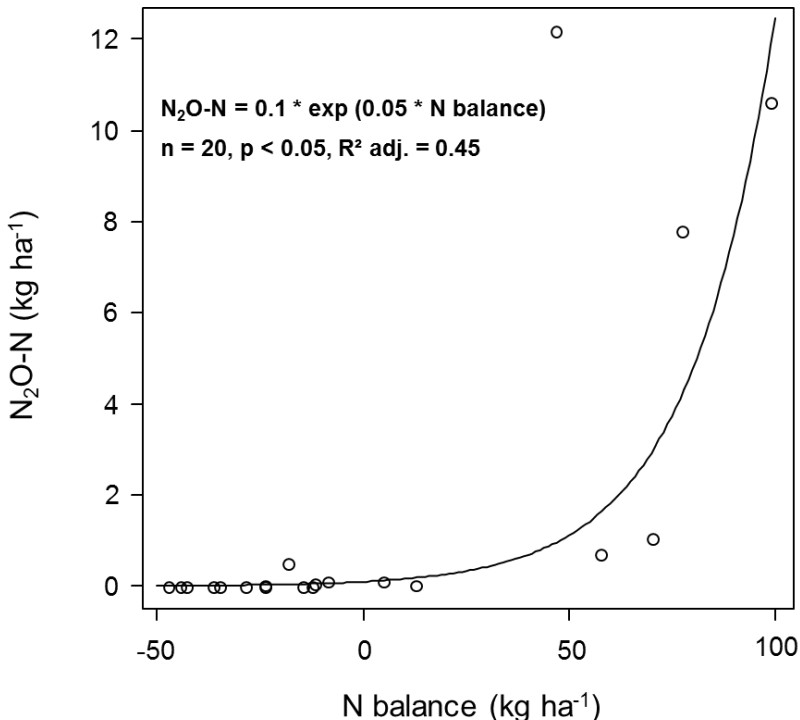

Figure 9. Relationship of cumulated $N_2O$ fluxes ($n = 8$) for a certain period of the growing seasons 2012
and 2013 at the arable site (AR) to nitrogen balance ($n = 3$) for the same period, calculated from mineral
N input of mineral and organic fertilizers and the N removal by plants. $R^2$ adjusted was estimated for
predicted versus obtained values.



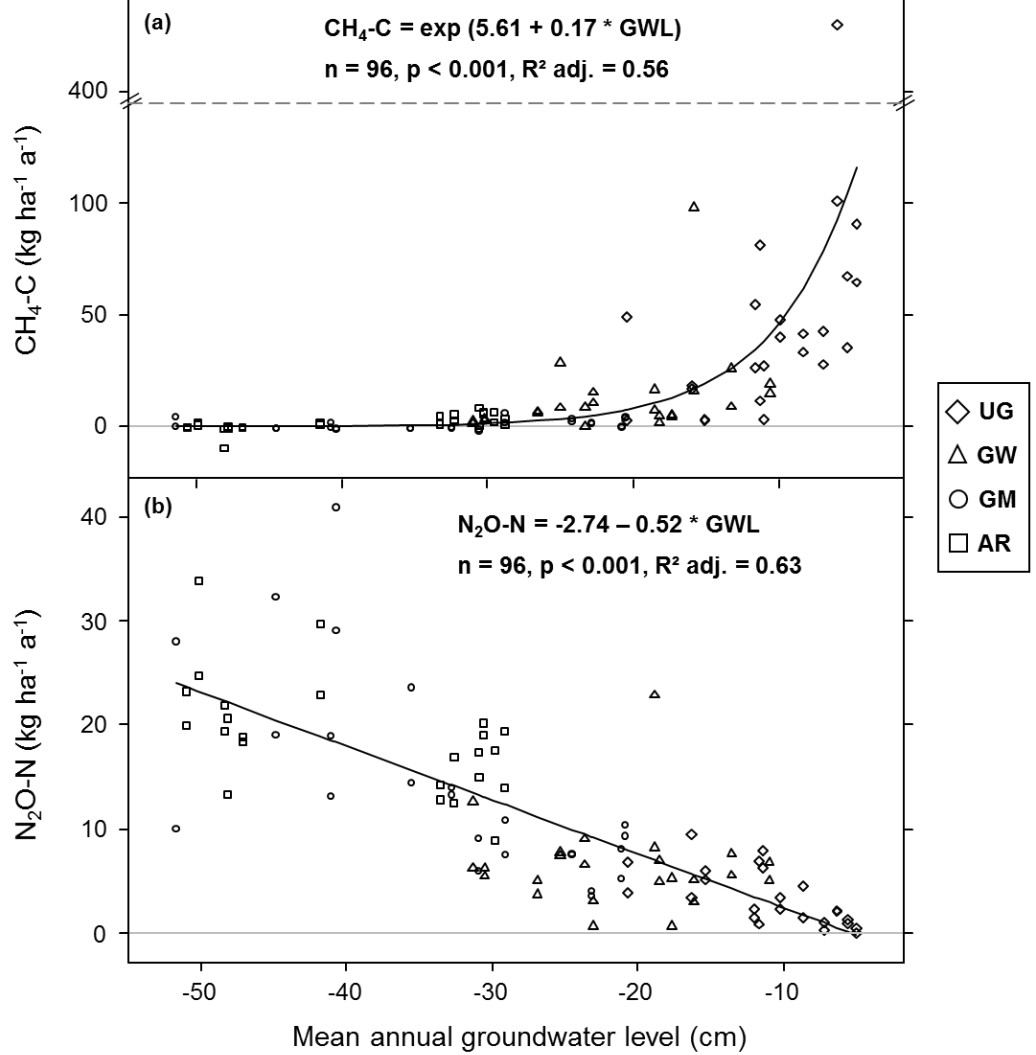

Figure 10. Relationships of cumulated annual $CH_4$ (a) and $N_2O$ fluxes (b) to mean annual groundwater
level for the study period (April 2011 – March 2014) with n = 8 per site and year. GWL in the equations
is mean annual groundwater level (cm). $R^2$ adjusted for exponential regression (a) was estimated for
predicted versus obtained values. Note the broken y-axis for figure (a).



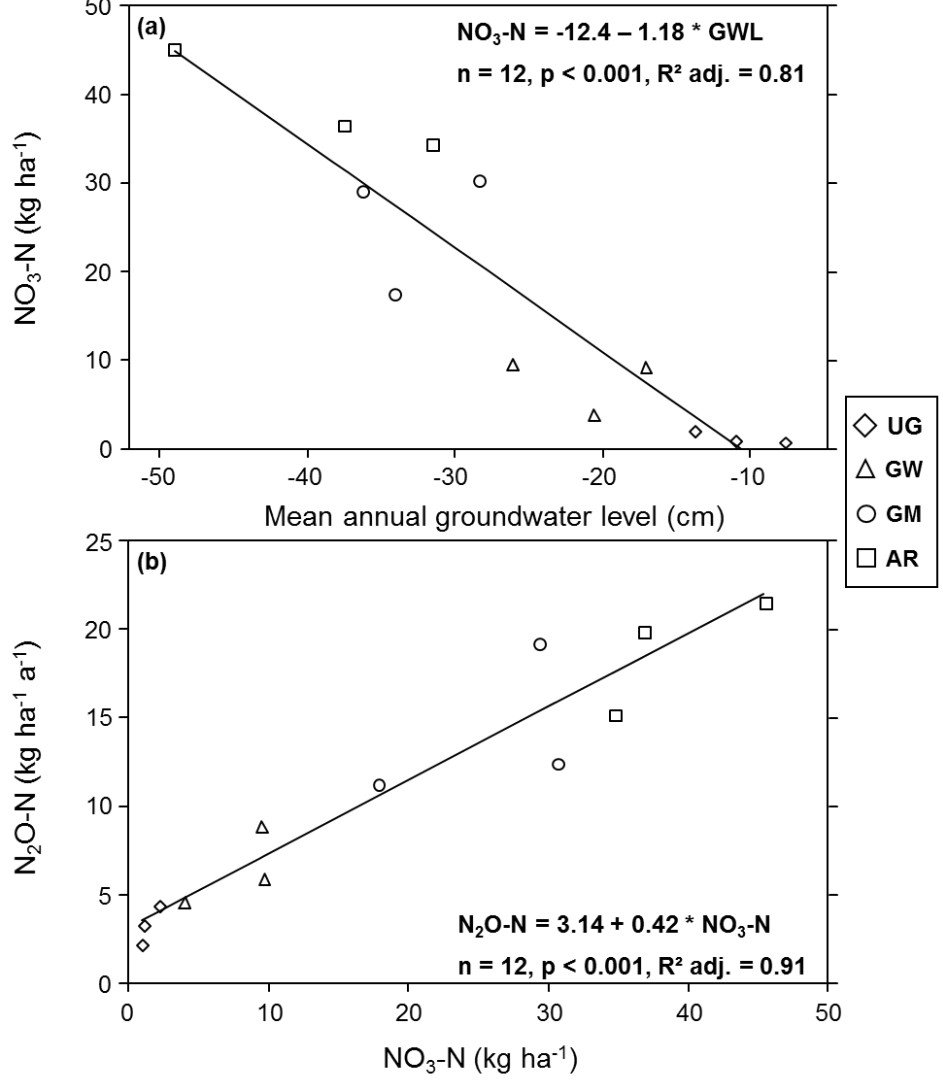

Figure 11. Relationship of mean annual amount of nitrate in 0 – 20 cm soil depth to mean annual
groundwater level (a) and relationship of mean annual $N_2O$ balances to mean annual amount of nitrate
in 0 – 20 cm soil depth (b). GWL in the equation (a) is mean annual groundwater level (cm), $NO_3$-N in
(b) is mean annual amount of nitrate in 0 – 20 cm (kg N ha$^{-1}$).



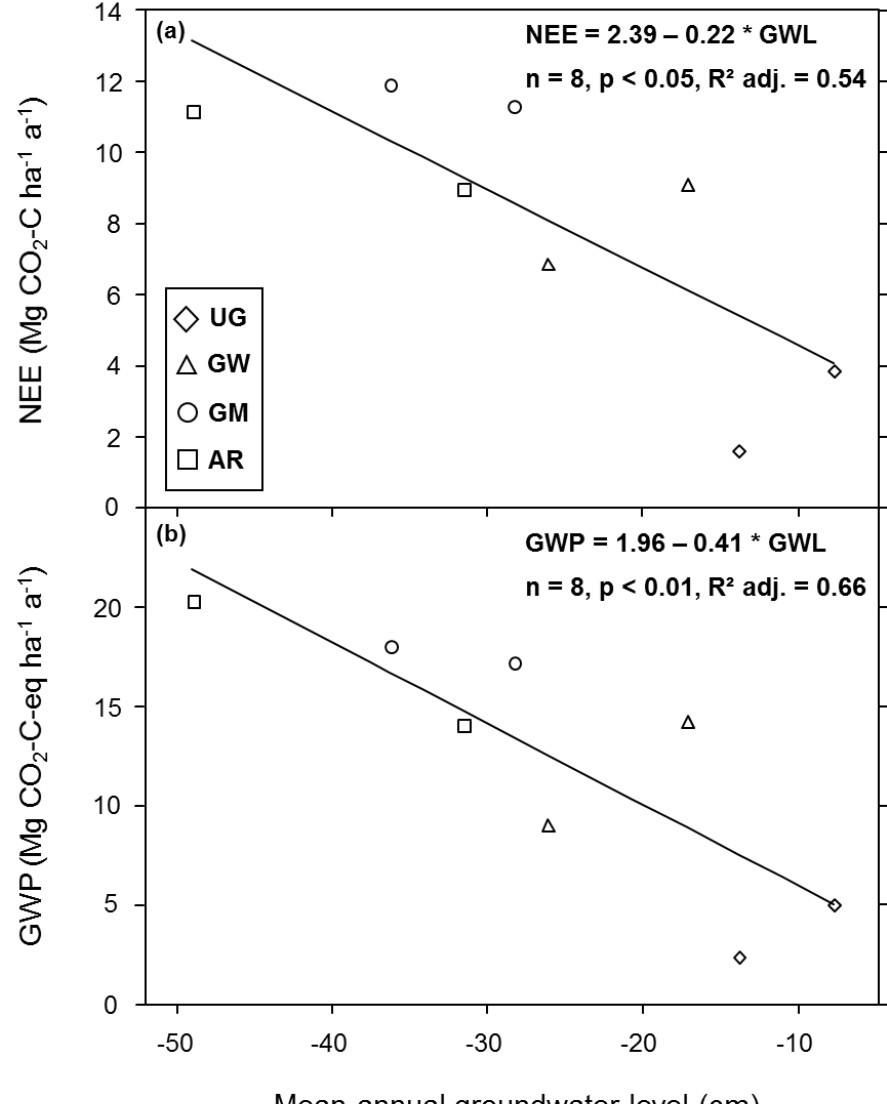

3    Figure 12. Effect of mean annual groundwater level on net ecosystem exchange (NEE) of $CO_2$ (a) and

4    global warming potential (GWP) of the four study sites (b) during the period April 2012 – March 2014.

5    GWL in the equations is mean annual groundwater level (cm).