# Peer review of "Greenhouse gas emissions from fen soils used for forage"

_Biogeosciences, 2015_

## Referee Comment (RC1) · Anonymous Referee #3 · 25 May 2016

This manuscript presents CO2, CH4 and N2O fluxes measured at four different field sites which are former peatlands and now managed and drained in different intensity. Greenhouse gas fluxes are presented over the year as well as annual balances. Multi-year measurements were carried out using the closed chamber method. The frequency of the measurements was weekly which is quite high for manual measurements and many additional measurements on soil properties were conducted as well. The measurement and modelling methods used in this study are well described. The topic of the manuscript is well within the scope of the journal. This paper is of high scientific quality, it is well structured. The results are presented in a clear way and discussion is comprehensive. I really like the figures, they are excellent. Only Figure 8 might need

some improvement as it is quite difficult to read maybe you could try to use different symbols than the filled dots. I really like the comparison of the carbon footprint of the different management practices. I highly recommend that manuscript to be published in Biogeosciences Discussions.

---

## Referee Comment (RC2) · Debjani Sihi (Referee) · 11 Aug 2016

Major comments

Poyda et al. addressed a very interesting question that how management and drainage intensity influences the greenhouse gas emissions from fen soils of northern Germany that are used for forage production. Research questions are important with respect to the current agricultural practices in this area. This is a timely study in light of the projected climate change and the importance of northern peatlands as a potential source or sink of atmospheric C.

The methods used for flux measurements and modeling are standard, statistical part

is described quite well. The study was done for three growing seasons, which helped to interpret the interannual variability of GHGs flux data. I like the idea of calculating the yield-related GHGs fluxes. In general, they reported that most important controller for GHGs emission in this system was long-term drainage intensity. Rewetting of this peat-based system reduced the overall global warming potential, but, still, the unutilized grassland (UG) act as a weak C source and a considerable amount of N2O was emitted to the atmosphere from UG site, and, highlighting the need to continue the effort to restore to the natural condition. The finding that the wet grassland has a potential for significant reduction in GHGs emission without removing the ongoing cultivation practices of forage crop should be of broad interest.

Specific comments and technical corrections:

P5, L22: "agriculturally" instead of "agricultural".

P7, L12: Place comma after "In these areas".

P8, L24: How long the oven drying of soil was done to estimate the gravimetric water content. It usually takes 24 hours at 105°C, but it is better to mention the duration.

P8, L25-28: How did you calculate organic C content from the elemental analyzer? The combustion method used in the elemental analyzer usually gives an estimate of total C. You can do acid digestion prior to the combustion step in order to eliminate the inorganic C or offline calculation can be done for organic C. If the assumption is that the peatland is mostly organic soil, then the estimates of loss on ignition are warranted. Please clarify this issue. Also, mention the time taken for oven drying of the samples at 40°C.

P9, L29: Is it normal to collect flux measurements between 9am to noon in these areas? When do you expect to see the peak in the diel pattern of CO2? Peak in the CO2 flux often lags by few hours with respect to the peak in the soil temperature in temperate and boreal forests (see Gaumont-Guay et al., 2006 and Savage et al.,

2009) due to a delayed response to the aboveground processes. The time lag in the agricultural system may be much less, if any. Also, to capture the daily mean value, you should take representative readings before and after the flux value peaked. An explanation on this may be of worth to support the sampling time used in this study for a representative mean daily flux.

P11, L8: What were your criteria for the acceptance of the $CO_2$ flux data? Did you follow the same approach like $CH_4$ and $N_2O$ flux data for the coefficient of determination?

P12, L23: What do you mean by "own examination"? Please explain briefly.

P13, L1-23: The equation for NEE is little confusing for the general reader. I see that you have mentioned the sign convention of individual flux components in L 19-21. But, it is better to write NEE=GPP-R_ECO and rephrase in the previous line that NEE was calculated as the difference (not sum) between GPP and R_ECO. It is better to maintain the conventional sign of flux: positive flux as a source to the atmosphere (which R_ECO is) and negative flux as a sink to the ecosystem (which GPP is) and the net balance of these two ultimately determine whether the ecosystem serves as a source (positive NEE) or a sink (negative NEE) for $CO_2$.

P13, L27: It will be worth exploring if the Kolmogorov-Smirnov Goodness-of-Fit Test for normality of data corroborates with the graphical residual analysis, especially for $CH_4$ and $N_2O$, which are often characterized by hot-spots or hot moments.

P22, L3-10: In general, fluxes of $N_2O$ (and $CO_2$) have been reported in literature after thawing of frozen soil due to the release of stored labile C and nutrients. The buildup of these labile substrates during freezing event usually comprised of dead microorganisms, dead fine roots, and C released from the breakdown of aggregates. Also, the response often depends on the intensity and duration of freezing as well as the soil properties. So, please explain clearly your point on the pulse of $N_2O$ during the freezing event, in addition to the thawing event afterward? Also, it has been reported that

the successive pulse of N2O has been reduced with increased frequency of freeze-thaw cycle, which may explain the lower winter fluxes in the second year. See Xu et al. (2016) and the articles cited in the reference list for more details.

P22, L11-20: Please note the prerequisite for the release of N2O in the incomplete denitrification process (where, complete denitrification: NO3- -> NO2- -> NO -> N2O -> N2) is the onset of anoxic (or reduced) condition. Do you have evidence that the N2O emission was greatest from nitrate-rich soils (or soil microsite) with relatively greater water filled pore space?

P24, L2: Check sentence. Consider "... would, therefore, ... before increase the total productivity..."

P24, L8: Consider "... source or sink of CO2..."

P28, L27: Place comma after "Also".

P53, Fig. 8: R2 adj for the model is 0.05. Does this mean ground water level and soil temperature at 5cm depth could explain only 5% of the variation in the flux of CH4-C? Please clarify the relevance of the model and what is the interpretation of this figure.

References cited in the review of the article that are not listed in the manuscript: 1. Gaumont-Guay, D., Black, T.A., Barr, A.G., Jassal, R.S., Nesic, Z., 2008. Biophysical controls on rhizospheric and heterotrophic components of soil respiration in a boreal black spruce stand. Tree Physiology 28, 161–171. 2. Savage, K., Davidson, E.A., Richardson, A.D. and Hollinger, D.Y., 2009. Three scales of temporal resolution from automated soil respiration measurements. Agricultural and Forest Meteorology, 149, 2012-2021. 3. Xu, X., Duan, C., Wu, H., Li, T., & Cheng, W. 2016. Effect of intensity and duration of freezing on soil microbial biomass, extractable C and N pools, and N2O and CO2 emissions from forest soils in cold temperate region.Science China Earth Sciences, 59, 156-169.

---

## Author Comment (AC1) · 15 Aug 2016

RC: How long the oven drying of soil was done to estimate the gravimetric water content? It usually takes 24 hours at 105 °C, but it is better to mention the duration.

AC: The soil samples were dried at 105 °C for roughly 24 hours. At the beginning of our study we tested if the sample weight would be further reduced during a second day of drying but we found that this was not the case.

RC: How did you calculate organic C content from the elemental analyzer? The combustion method used in the elemental analyzer usually gives an estimate of total C. You can do acid digestion prior to the combustion step in order to eliminate the inorganic C

or offline calculation can be done for organic C. If the assumption is that the peatland is mostly organic soil, then the estimates of loss on ignition are warranted. Please clarify this issue. Also, mention the time taken for oven drying of the samples at 40 °C.

AC: As the peat soil in our study was free of inorganic C, the total C determined by the combustion method equals the organic C content of the soil sample. Oven drying at 40 °C was done for 2 – 4 days, depending on the moisture content of the samples.

RC: Is it normal to collect flux measurements between 9am to noon in these areas? When do you expect to see the peak in the diel pattern of CO2? Peak in the CO2 flux often lags by few hours with respect to the peak in the soil temperature in temperate and boreal forests (see Gaumont-Guay et al., 2006 and Savage et al., 2009) due to a delayed response to the aboveground processes. The time lag in the agricultural system may be much less, if any. Also, to capture the daily mean value, you should take representative readings before and after the flux value peaked. An explanation on this may be of worth to support the sampling time used in this study for a representative mean daily flux.

AC: During the mentioned period of the day we expected the mean daily fluxes of N2O and CH4 as it was described in other studies. For example, van der Weerden et al. (2013) conducted near-continuous measurements of N2O fluxes with automatic chambers and stated that mean daily fluxes occurred between 10:00 and 12:00 h and 18:00 and 21:00 h. We expected the peak GPP at the same time when PAR is at its maximum, which is generally between 12:00 and 13:00 h. Maximum daily RECO is usually measured few hours later due to the time lack of temperature maxima. As RECO is the sum of autotrophic and heterotrophic respiration, it depends on both, air and soil temperature. We did not expect a time lag between the maxima of these temperatures and the respiration. However, our CO2 measurement campaigns were conducted from sunrise until the afternoon (approximately 4 h after PAR peaked) to cover the daily range of radiation and temperature and thus assimilation and respiration. We usually stopped our daily campaigns when we saw no increase in the RECO flux anymore.

RC: What were your criteria for the acceptance of the CO2 flux data? Did you follow the same approach like CH4 and N2O flux data for the coefficient of determination?

AC: Quality criteria for CO2 measurements were changes of chamber temperature of more than 1.5 °C and a standard deviation of PAR more than 10 % of average PAR for NEE measurements (transparent chamber). If these thresholds were exceeded during a measurement, the CO2 flux was not used for further analyses. Additionally, each single CO2 measurement was carefully checked and the flux was only calculated for that part of the measurement with a linear concentration change over time. The $R^2$ was not used as a quality criteria for the CO2 fluxes as there were up to 60 data points (CO2 concentrations) during one measurement and especially for measurements in winter times when fluxes were very low, a $R^2$ of $\geq 0.9$ could hardly be fulfilled.

RC: What do you mean by "own examination"? Please explain briefly.

AC: I measured the PAR inside and outside the chamber at different light intensities and found that PAR inside the chamber was on average 8 % lower than outside the chamber.

RC: The equation for NEE is little confusing for the general reader. I see that you have mentioned the sign convention of individual flux components in L 19-21. But, it is better to write NEE=GPP - R_ECO and rephrase in the previous line that NEE was calculated as the difference (not sum) between GPP and R_ECO. It is better to maintain the conventional sign of flux: positive flux as a source to the atmosphere (which R_ECO is) and negative flux as a sink to the ecosystem (which GPP is) and the net balance of these two ultimately determine whether the ecosystem serves as a source (positive NEE) or a sink (negative NEE) for CO2.

AC: We applied the conventional sign convention just like you mentioned it. As GPP is negative, RECO has to be added to get the NEE. In recent studies observing the NEE of peatland ecosystems, the equation is written as a sum of the two processes and we wanted to be consistent at this point.

RC: It will be worth exploring if the Kolmogorov-Smirnov Goodness-of-Fit Test for normality of data corroborates with the graphical residual analysis, especially for CH4 and N2O, which are often characterized by hot-spots or hot moments.

AC: As we apply the graphical residual analysis as a standard procedure to all data sets as a decision support tool for the statistical analyses, it was done in the same way with the CH4 and N2O flux data. Additional tests could be conducted as well but have not been designated due to consistency issues.

RC: In general, fluxes of N2O (and CO2) have been reported in literature after thawing of frozen soil due to the release of stored labile C and nutrients. The buildup of these labile substrates during freezing event usually comprised of dead microorganisms, dead fine roots, and C released from the breakdown of aggregates. Also, the response often depends on the intensity and duration of freezing as well as the soil properties. So, please explain clearly your point on the pulse of N2O during the freezing event, in addition to the thawing event afterward? Also, it has been reported that the successive pulse of N2O has been reduced with increased frequency of freeze-thaw cycle, which may explain the lower winter fluxes in the second year. See Xu et al. (2016) and the articles cited in the reference list for more details.

AC: Our results show that the mentioned N2O pulse is occurring during freezing events but N2O fluxes decline rapidly after freezing. This was more pronounced when no snow cover was present. These two points suggest that the predominating process that enhanced winter N2O fluxes was the freezing rather than the thawing of the peat soils. As the N2O flux did not increase directly after air temperatures became negative but a few days afterwards, underlines this conclusion as the wet peat soils have a high heat capacity, which means that the time lag between changes in air and soil temperatures is greater. This could explain the missing N2O pulse in the second winter as the frost could not penetrate the peat sufficiently to generate an enhanced release of C and N. However, due to the very high amounts of C stored in the peat soils and the densely rooted top soils in combination with high nutrient loads on the agricultural sites, the

investigated peatlands have a high potential for N2O emissions in general and also for freezing induced N2O pulses.

RC: Please note the prerequisite for the release of N2O in the incomplete denitrification process (where, complete denitrification: NO3- -> NO2- -> NO -> N2O -> N2) is the onset of anoxic (or reduced) condition. Do you have evidence that the N2O emission was greatest from nitrate-rich soils (or soil microsite) with relatively greater water filled pore space?

AC: N2O emissions were not significantly related to water filled pore space. The ground water level (GWL) was the dominating factor for annual N2O emissions at our sites with increasing emissions at lower mean annual GWL. However, it has to be noted that with lower mean annual GWL also the fluctuations of the GWL are increasing, which means that there is a thicker active peat layer where N can be mineralized and nitrified. The produced nitrate that is dislocated to saturated pores will then be denitrified with potential losses of N2O. As a consequence, daily N2O fluxes of peat soils can hardly be related to the GWL or water filled pore space at a certain day, at least in field studies. We only found a significant relation between the daily fluxes and the amount of nitrate in the topsoil as the occurrence of high amounts of nitrate that exceed plant uptake can lead to incomplete denitrification and thus N2O release.

RC: R2 adj for the model is 0.05. Does this mean ground water level and soil temperature at 5 cm depth could explain only 5% of the variation in the flux of CH4-C? Please clarify the relevance of the model and what is the interpretation of this figure.

AC: The figure is not an illustration of the model as we used a multiple linear regression model with log-transformed daily CH4 fluxes where the site was used as a covariate, additionally to the groundwater level and soil temperature at 5 cm depth. The figure was made to illustrate the extremely high variability of CH4 fluxes between the sites but also within single sites and to show that highest emissions occurred when both groundwater level and soil temperature were high. However, this was highly depending

on the location as the deep-drained sites showed negligible fluxes irrespective of GWL and soil temperature. This was underlined by the model as all three covariates had a highly significant effect on CH4 fluxes.

———————————————————————

---

## Author Comment (AC2) · 15 Aug 2016

I very much appreciate the positive evaluation of the manuscript. The mentioned figure 8 will be improved to make it easier to read.

---

## Author Response (AR1)

**Response to the reviews**

- **How long the oven drying of soil was done to estimate the gravimetric water content? It usually takes 24 hours at 105 °C, but it is better to mention the duration.**

The soil samples were dried at 105 °C for roughly 24 hours. At the beginning of our study we tested if the sample weight would be further reduced during a second day of drying but we found that this was not the case.

- **How did you calculate organic C content from the elemental analyzer? The combustion method used in the elemental analyzer usually gives an estimate of total C. You can do acid digestion prior to the combustion step in order to eliminate the inorganic C or offline calculation can be done for organic C. If the assumption is that the peatland is mostly organic soil, then the estimates of loss on ignition are warranted. Please clarify this issue. Also, mention the time taken for oven drying of the samples at 40 °C.**

As the peat soil in our study was free of inorganic C, the total C determined by the combustion method equals the organic C content of the soil sample. Oven drying at 40 °C was done for 2 – 4 days, depending on the moisture content of the samples.

- **Is it normal to collect flux measurements between 9am to noon in these areas? When do you expect to see the peak in the diel pattern of $CO_2$? Peak in the $CO_2$ flux often lags by few hours with respect to the peak in the soil temperature in temperate and boreal forests (see Gaumont-Guay et al., 2006 and Savage et al., 2009) due to a delayed response to the aboveground processes. The time lag in the agricultural system may be much less, if any. Also, to capture the daily mean value, you should take representative readings before and after the flux value peaked. An explanation on this may be of worth to support the sampling time used in this study for a representative mean daily flux.**

During the mentioned period of the day we expected the mean daily fluxes of $N_2O$ and $CH_4$ as it was described in other studies. For example, van der Weerden et al. (2013) conducted near-continuous measurements of $N_2O$ fluxes with automatic chambers and stated that mean daily fluxes occurred between 10:00 and 12:00 h and 18:00 and 21:00 h.

We expected the peak GPP at the same time when PAR is at its maximum, which is generally between 12:00 and 13:00 h. Maximum daily $R_{ECO}$ is usually measured few hours later due to the time lag of temperature maxima. As $R_{ECO}$ is the sum of autotrophic and heterotrophic respiration, it depends on both, air and soil temperature. We did not expect a time lag between the maxima of these temperatures and the respiration. However, our $CO_2$ measurement campaigns were conducted from sunrise until the afternoon (approximately 4 h after PAR peaked) to cover the daily range of radiation and temperature and thus assimilation and respiration. We usually stopped our daily campaigns when we saw no increase in the $R_{ECO}$ flux anymore.

- **What were your criteria for the acceptance of the $CO_2$ flux data? Did you follow the same approach like $CH_4$ and $N_2O$ flux data for the coefficient of determination?**

Quality criteria for $CO_2$ measurements were changes of chamber temperature of more than 1.5 °C and a standard deviation of PAR more than 10 % of average PAR for NEE measurements (transparent chamber). If these thresholds were exceeded during a measurement, the $CO_2$ flux was not used for further analyses. Additionally, each single $CO_2$ measurement was carefully checked and the flux was only calculated for that part of the measurement with a linear concentration change over time. The R² was not used as a quality criteria for the $CO_2$ fluxes as there were up to 60 data points ($CO_2$ concentrations) during one measurement and especially for measurements in winter times when fluxes were very low, a R² of ≥ 0.9 could hardly be fulfilled.

- **What do you mean by "own examination"? Please explain briefly.**

I measured the PAR inside and outside the chamber at different light intensities and found that PAR inside the chamber was on average 8 % lower than outside the chamber.

- **The equation for NEE is little confusing for the general reader. I see that you have mentioned the sign convention of individual flux components in L 19-21. But, it is better to write NEE=GPP - R_ECO and rephrase in the previous line that NEE was calculated as the difference (not sum) between GPP and R_ECO. It is better to maintain the conventional sign of flux: positive flux as a source to the atmosphere (which R_ECO is) and negative flux as a sink to the ecosystem (which GPP is) and the net balance of these two ultimately determine whether the ecosystem serves as a source (positive NEE) or a sink (negative NEE) for $CO_2$.**

We applied the conventional sign convention just like you mentioned it. As GPP is negative, $R_{ECO}$ has to be added to get the NEE. In recent studies observing the NEE of peatland ecosystems, the equation is written as a sum of the two processes and we wanted to be consistent at this point.

- **It will be worth exploring if the Kolmogorov-Smirnov Goodness-of-Fit Test for normality of data corroborates with the graphical residual analysis, especially for $CH_4$ and $N_2O$, which are often characterized by hot-spots or hot moments.**

As we apply the graphical residual analysis as a standard procedure to all data sets as a decision support tool for the statistical analyses, it was done in the same way with the $CH_4$ and $N_2O$ flux data. Pre-tests are often not recommended any more (see Rasch et al., 2011). An important reason is that pre-tests such as the Kolmogorov-Smirnov-Test only indicate if the data are not significantly normal distributed but they cannot give evidence about normal distribution which we would need to know. Therefore, we used the graphical residual analysis to determine the distribution of data.

- **In general, fluxes of $N_2O$ (and $CO_2$) have been reported in literature after thawing of frozen soil due to the release of stored labile C and nutrients. The buildup of these labile substrates during freezing event usually comprised of dead microorganisms, dead fine roots, and C released from the breakdown of aggregates. Also, the response often depends on the intensity and duration of freezing as well as the soil properties. So, please explain clearly your point on the pulse of $N_2O$ during the freezing event, in addition to the thawing event afterward? Also, it has been reported that the successive pulse of $N_2O$ has been reduced with increased frequency of freeze-thaw cycle, which may explain the lower winter fluxes in the second year. See Xu et al. (2016) and the articles cited in the reference list for more details.**

Our results show that the mentioned $N_2O$ pulse is occurring during freezing events but $N_2O$ fluxes decline rapidly after freezing. This was more pronounced when no snow cover was present. These two points suggest that the predominating process that enhanced winter $N_2O$ fluxes was the freezing rather than the thawing of the peat soils. As the $N_2O$ flux did not increase directly after air temperatures became negative but a few days afterwards, underlines this conclusion as the wet peat soils have a high heat capacity, which means that the time lag between changes in air and soil temperatures is relatively great. This could explain the missing $N_2O$ pulse in the second winter as the frost could not penetrate the peat sufficiently to generate an enhanced release of C and N. However, due to the very high amounts of C stored in the peat soils and the densely rooted top soils in combination with high nutrient loads on the agricultural sites, the investigated peatlands have a high potential for $N_2O$ emissions in general and also for freezing induced $N_2O$ pulses.

- **Please note the prerequisite for the release of $N_2O$ in the incomplete denitrification process (where, complete denitrification: $NO_3^- \rightarrow NO_2^- \rightarrow NO \rightarrow N_2O \rightarrow N_2$) is the**

**onset of anoxic (or reduced) condition. Do you have evidence that the $N_2O$ emission was greatest from nitrate-rich soils (or soil microsite) with relatively greater water filled pore space?**

$N_2O$ emissions were not significantly related to water filled pore space. The ground water level (GWL) was the dominating factor for annual $N_2O$ emissions at our sites with increasing emissions at lower mean annual GWL. However, it has to be noted that with lower mean annual GWL also the fluctuations of the GWL are increasing, which means that there is a thicker active peat layer where N can be mineralized and nitrified. The produced nitrate that is dislocated to saturated pores will then be denitrified with potential losses of $N_2O$. As a consequence, daily $N_2O$ fluxes of peat soils can hardly be related to the GWL or water filled pore space at a certain day, at least in field studies. We only found a significant relation between the daily fluxes and the amount of nitrate in the topsoil as the occurrence of high amounts of nitrate that exceed plant uptake can lead to incomplete denitrification and thus $N_2O$ release.

- **$R^2$ adj for the model is 0.05. Does this mean ground water level and soil temperature at 5 cm depth could explain only 5% of the variation in the flux of $CH_4$-C? Please clarify the relevance of the model and what is the interpretation of this figure.**

The figure is not an illustration of the model as we used a multiple linear regression model with log-transformed daily $CH_4$ fluxes where the site was used as a covariate, additionally to the groundwater level and soil temperature at 5 cm depth. The figure was made to illustrate the extremely high variability of $CH_4$ fluxes between the sites but also within single sites and to show that highest emissions occurred when both groundwater level and soil temperature were high. However, this was highly depending on the location as the deep-drained sites showed negligible fluxes irrespective of GWL and soil temperature. This was underlined by the model as all three covariates had a highly significant effect on $CH_4$ fluxes. In addition, the model comprised additional terms as for example the year as a random factor and a heteroscedasticity term due to site and year. However, this model only explained 11 % of the variation in the $CH_4$ flux. The $R^2$ adj. in Fig. 8 indicates that only 5 % are explained when only the GWL and the soil temperature are considered as influencing factors.

**List of changes**

- **Sec. 2.2.2:** The duration of oven drying at 105 °C (p10, l 23) and at 40 °C (p10, l 25) as well as a note on the determination of organic C content (L 26 – 28) has been added.
- **Sec. 2.3.1:** Notes on the time of gas sampling (p 11, l 28 – 29) and $CO_2$ measurements (p 12, l 18 – 20) have been added.
  The last sentence has been removed (p 12, l 21 – 23).
- **Sec. 2.3.2:** The explanation of quality criteria for $CO_2$ flux measurements has been added (p 13, l 14 -16).
- **Sec. 2.3.3:** The determination of PAR absorption by transparent chambers is explained in more detail (p 14, l 19 – 21).
- **Sec. 2.4:** A reference has been added (p 15, l 24 – 25).
- **Sec. 4.1:** A note on the $CH_4$ model's explanatory power was added, including a citation (p 20, l 27 – 29).
- **Sec. 4.2:** A note on the $N_2O$ model's explanatory power was added, including a citation (p 22, l 12 – 14).
  $N_2O$ fluxes during freeze-thaw events were discussed in more detail (p 23, l 31 – p 24, l 10).
- **References:** Additional references were added:
  Nakagawa & Schielzeth (2013) (p 39, l 1 – 3)
  Rasch et al. (2011) (p 39, l 26 – 27)
  Xu et al. (2016) (p 43, 4 – 6)

[revised manuscript text omitted]